# The European summer heatwave 2019 - a regional storyline perspective

Tatiana Klimiuk[1], Patrick Ludwig[1], Antonio Sanchez-Benitez[2], Helge F. Goessling[2], Peter Braesicke[3], and Joaquim G. Pinto[1]

[1]Institute of Meteorology and Climate Research - Tropospheric Research (IMKTRO), Karlsruhe Institute of Technology (KIT), Karlsruhe, Germany
[2]Alfred Wegener Institute Helmholtz-Center for Polar and Marine Research, Bremerhaven, Germany
[3]Institute of Meteorology and Climate Research - Atmospheric Trace Gases and Remote Sensing (IMKASF), Karlsruhe Institute of Technology (KIT), Karlsruhe, Germany

**Correspondence:** Tatiana Klimiuk (tatiana.klimiuk@kit.edu)

**Abstract.**

The number and intensity of heat waves have increased in the recent past, along with anthropogenic climate change. This poses challenges to many communities and raises the need to develop adaptation measures based on more accurate information regarding regional to local changes in temperature extremes and their impacts. While the general increase in global mean temperature is well established, current global climate projections show a large model spread regarding possible future circulation changes. To isolate the more certain thermodynamic response from the less certain dynamical response to anthropogenic climate change, we employ an event-based storyline approach and focus the present study on the 2019 summer heatwaves over Central Europe. Our approach comprises three steps. Firstly, the large-scale circulation in the free troposphere was spectrally nudged to the ERA5-reanalyses in the global coupled climate model AWI-CM-1.1-MR for a recent period (2017 - 2022), corresponding to +1.4 K global warming, and repeated under pre-industrial, +2 K, +3 K, and +4 K global warming climates. Secondly, the global storylines were dynamically downscaled with the regional ICON-CLM model to the Euro-CORDEX domain with a horizontal resolution of 12 km and, thirdly, to a Central-European (German) domain with a resolution of 3 km. We provide evidence that the downscaling of global storyline integrations significantly improved the representation of present-day temperature patterns and reduced error in daily 2m temperature relative to observations in Central Europe. The magnitude of the heatwave temperature response significantly exceeds the globally modelled background warming, with a distinct spatial and temporal variation in the regional increments. Our simulations indicate a general linear dependency of the 2m temperature response to the global warming level: the warming rates during the July 2019 heatwave ranged between factors of 2 and 3 in Central Europe, resulting in an anthropogenic warming of 8 to 12 °C in the +4 K climate. The spatial extent and the duration of the heat wave are also amplified in the warmer climates. With this three-step downscaling approach, we gain new insights into possible future changes in heat extremes in Central Europe, which apparently surpass global warming trends. Along with its scientific value, our method provides ways to facilitate communication of regional climate change information to the users.

# 1   Introduction

Heat waves are a major natural hazard worldwide, with the heat waves of 2003, 2010 and 2018 as prominent examples on the
European continent (e.g., Fink et al., 2004; Barriopedro et al., 2011; Miralles et al., 2014; Spensberger et al., 2020). In the
last two decades, Europe has witnessed an increase in the frequency, duration, and intensity of extreme heat events, which in
turn is causing an increase in mortality rates, food and water insecurity, and long-term economic and cultural stress (Robine
et al., 2008; García-Herrera et al., 2010; Perkins-Kirkpatrick and Lewis, 2020; Becker et al., 2022; Calvin et al., 2023; Knutzen
et al., 2023). The general effects of climate change on heat wave characteristics have been demonstrated to be robust in recent
studies that are usually aimed at estimating the effect of human-induced greenhouse gas (GHG) forcing to the recent extreme
events using observational records and at projecting future changes by means of climate modelling (e.g., Barriopedro et al.,
2011, 2023). The traditional approaches that commonly imply multi-model averaging and probabilistic event attribution can
provide an estimate of trends in frequency, intensity, and persistence of extreme events, but they lack clarity regarding the
physical processes that are causing the changes (Shepherd, 2014).

One of the current scientific challenges is to disentangle the relative role of dynamical and thermodynamic contributions
to future heat wave characteristics in attribution and projection studies (Shepherd, 2014, 2021; Sousa et al., 2020; Sánchez-
Benítez et al., 2018). Thermodynamic effects, such as near-surface warming, moistening of the atmosphere, and effects from
the partitioning of radiative and turbulent fluxes, show a relatively robust response to the anthropogenic GHG forcing in models
and generally tend to have less internal variability (Deser et al., 2014; Wehrli et al., 2018). On the other hand, the dynamic
effects that include changes in the position, strength and meandering of the jet stream, as well as changes in the occurrence of
weather regimes and more local circulation patterns, are subject to larger uncertainties (Deser et al., 2014; Shepherd et al., 2018;
Zappa, 2019). First, this is associated with the inherent model uncertainties and differences in parametrisations of unresolved
processes in the models (Shepherd, 2014); second, the internal variability of the dynamical component of the atmosphere is
responsible for a low signal-to-noise ratio in the studies aimed to quantify the regional response of extreme events to global
warming (Deser et al., 2014; Shepherd, 2014, 2021; Wehrli et al., 2018; Barriopedro et al., 2023).

An alternative to circumvent the uncertainties associated with different atmospheric circulation changes under enhanced
GHG forcing is to consider an event-based storyline approach, where the dynamical conditions are constrained to the present-
day state in a specified way (Caviedes-Voullième and Shepherd, 2023). Along with the uncertainty of changing dynamics,
the internal variability is considerably reduced in event-based storylines (Sánchez-Benítez et al., 2022), which improves the
signal-to-noise ratio, giving us the opportunity to better quantify the actual event-specific thermodynamic response. Assuming
the robustness of the quantification, we improve our understanding of the potential impacts of future extreme events and
communicate them to the public and authorities in a more understandable manner.

Constraining the dynamical conditions on a regional scale can be achieved with the Pseudo Global Warming (PGW) ap-
proach, i.e., by perturbing the boundary conditions obtained from reanalyses with the average climate change signal (deltas)
from global climate models (GCMs) (Schär et al., 1996; Aalbers et al., 2023; Ludwig et al., 2023; Vries et al., 2024). As
only smoothed multi-year averaged GCM fields are used to modify the boundary conditions, this method is computationally

effective, which is a significant advantage for multi-model and multi-ensemble studies of regional thermodynamic response to global warming (Brogli et al., 2023).

While the inter-annual variability of delta fields obtained from GCMs is intentionally suppressed in the PGW approach, it can be consistently taken into account in spectrally nudged storylines. In this method, a GCM is run with the large-scale circulation forced to follow a reanalyses state by nudging the upper tropospheric winds, whereas the background climate corresponds to a specific warming level (Sánchez-Benítez et al., 2022; Athanase et al., 2024; van Garderen et al., 2021; Wehrli et al., 2020). If the GCM is a coupled model as in Sánchez-Benítez et al., 2022, no assumptions on the deltas of sea surface temperature and sea ice content have to be made (van Garderen et al., 2021).

On the other hand, one of the advantages of the PGW approach over the nudged storyline approach is the potential to avoid GCM-specific biases by repeating the experiment with deltas derived from multi-model ensemble means or different single-model multi-member ensemble means (see, e.g., Aalbers et al., 2023; Vries et al., 2024). In our work, we follow the path prescribed by a single GCM. In the context of the storyline approach, this unfolding of events is physically self-consistent and plausible, which complies with the definition of a storyline introduced by (Shepherd et al., 2018) and allows for a process-oriented evaluation of the obtained responses.

Nevertheless, GCMs have difficulty representing the regional climate mean and variability due to unresolved orography and shortcomings in model parameterisations associated with the coarse horizontal resolution (Giorgi and Gutowski Jr, 2015). For example, using global nudged storylines, Sánchez-Benítez et al. (2022) found a strong amplification for the July 2019 European heat wave under global warming; however, their global simulations underestimated the high-temperature extremes reached during the heatwave. An effective method to address the lack of precision in GCMs is to perform the dynamical downscaling with a regional climate model (RCM) (Feser et al., 2011; Giorgi, 2019; Vautard et al., 2021). Commonly, the horizontal resolution of RCMs applied for the European domain is in the order of 10 - 12 km (e.g., Jacob et al., 2014; Giorgi and Gutowski Jr, 2015). Being computationally efficient, this resolution allows for the production of large ensembles of simulations, which considerably improves the representation of relevant climatological variables compared to GCMs (Vautard et al., 2021). Nevertheless, convective processes can only be resolved by convection-permitting regional models (CPMs) operating at resolutions finer than 4 km (Prein et al., 2015; Giorgi, 2019; Hundhausen et al., 2023). Associated with explicitly resolved deep convection and better-captured processes in regions with complex topography, CPMs have been shown to add further value to RCM's representation of precipitation and near-surface temperature, especially in the regions with complex topography (Prein et al., 2015; Giorgi, 2019).

In this study, we go a step beyond the global storyline approach by providing a regional perspective of the heat wave that occurred in Europe during July 2019 and its unfolding in colder (pre-industrial) and warmer climates (see, e.g., Sánchez-Benítez et al., 2022; Sousa et al., 2020). With this aim, we dynamically downscale the global spectrally nudged storylines for the summer 2019 heatwave for five different background climates ranging from pre-industrial to +4 K global warming for (Central) Europe. The approach utilises a global-to-regional (GCM-RCM-CPM) model chain comprising the global spectrally nudged storyline simulations obtained from the global Alfred Wegener Institute coupled climate model AWI-CM-1.1-MR (hereafter referred to as AWI-CM1, Semmler et al., 2020) with the large-scale horizontal winds spectrally nudged to ERA5

(Hersbach et al., 2020), and the ICOsahedral Nonhydrostatic model in Climate Limited area Mode ICON-CLM (Pham et al., 2021) for the dynamical downscaling to 12 km horizontal grid spacing over Europe and subsequently to 3 km horizontal grid spacing over Central Europe. This approach permits the derivation of climate data at high resolution, thereby providing detailed information for attribution and impact studies.

95    We address the following research questions:

(1) How accurately can a regional event-based storyline simulation represent a recent event, and what is the improvement compared to the global spectrally nudged storyline simulation?

(2) What is the effect of climate change on the 2019 European heatwave based on the regional and convection-permitting ICON-CLM simulations?

(3) What is the local to regional extreme temperature scaling in response to global warming for an event like the 2019 heat wave?

The paper is structured as follows: Section 2 describes the global and regional model setups, as well as the datasets used for model evaluation. The main results are presented in Section 3, with the first research question addressed in Section 3.1 and the regional storylines analysed in Sections 3.2 and 3.3. Section 4 summarises and discusses the results, formulates the main 105    conclusions for each research question, and provides future research ideas.

## 2    Data and methods

### 2.1    Global Spectrally Nudged Storylines

The global spectrally nudged simulations are based on the global coupled climate model AWI-CM1 (Semmler et al., 2020). This model has contributed to phase 6 of the Coupled Model Intercomparison Project (CMIP6, Eyring et al., 2016). It consists 110    of the atmospheric model ECHAM6.3.04p1 from MPI-M (Stevens et al., 2013) coupled to the Finite Element Sea Ice–Ocean Model (FESOM) v.1.4 for the ocean component (Wang et al., 2014). The atmospheric component is run at a T127L95 spectral resolution, which corresponds to a horizontal resolution of about 100 km in the tropics and 95 vertical levels going up to ∼0.01 hPa. The ocean model FESOM uses an unstructured mesh that allows for fine resolution in energetically active areas such as the Gulf Stream (Sidorenko et al., 2015; Sein et al., 2017). Consequently, the horizontal resolution of the ocean ranges from 80 115    km in the subtropical Pacific to 8-10 km in the North Sea and 8-20 km in the Arctic (see Fig. 1 in Semmler et al., 2020).

In the storyline experiments, the evolution of the AWI-CM1 large-scale atmospheric circulation is constrained by spectrally nudging the model's vorticity and divergence (representing the large-scale horizontal winds) to ERA5 reanalysis data (Hersbach et al., 2020) with an e-folding time of 24 h and a spectral truncation of 20 on zonal wavenumbers. Nudging is applied only to vertical levels between 700 and 100 hPa. This configuration has been shown to optimally constrain large-scale events such as 120    heatwaves (Sánchez-Benítez et al., 2022), warm and moist intrusions in the Arctic (Pithan et al., 2023) or Marine heatwaves (Athanase et al., 2024), while preserving some freedom in the boundary layer and at small spatiotemporal timescales.

A series of nudged storyline simulations were conducted using ESM Tools (Barbi et al., 2021) for a range of climate states based on the configuration described above. Specifically, nudging experiments were branched off the historical CMIP6 runs

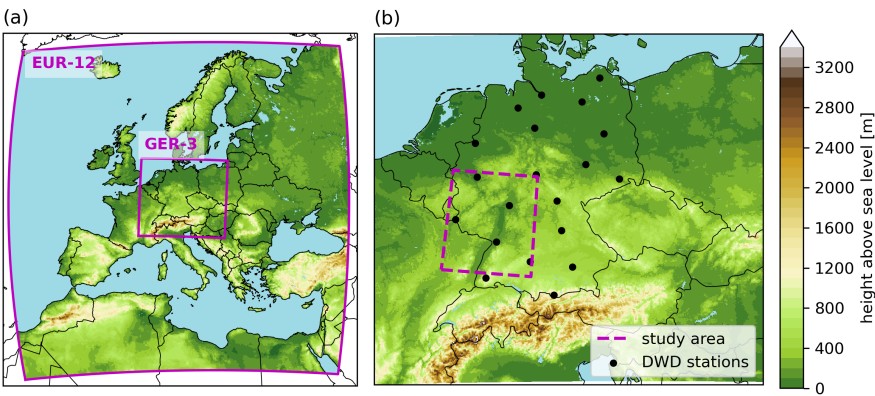

**Figure 1.** (a) EUR-12 and GER-3 domains used for downscaling the global AWI-CM1 storylines with ICON-CLM. (b) GER-3 domain with the locations of DWD (German Weather Service) observation stations and the area used for spatial averaging (48° N - 51° N, 6° E - 10° E). Shading corresponds to the respective orography used in the simulations. The shown domains include the lateral boundary zone.

(Semmler et al., 2020) on 1st of January 1851 to produce pre-industrial climate conditions. Meanwhile, present-day (+1.4 K)
and +2 K, +3 K, +4 K climates relative to pre-industrial were made by spawning the shared socioeconomic pathway scenario ssp370 CMIP6 experiments (Semmler et al., 2019) on 1st of January 2017, 2038, 2065, and 2093, respectively. These years were selected according to when the global warming levels were reached. All storylines comprise five ensemble members, each spawned from the five respective CMIP6 simulations and thus started from slightly different initial conditions. Each storyline is simulated continuously from the 1st of January 2017 to the 30th of September 2022. Throughout the text, for all storylines, we refer to the years corresponding to the present-day circulation inferred from ERA5 as the "dynamical years" 2017–2022.

## 2.2 Dynamical downscaling with ICON-CLM

Next, the data from the global spectrally nudged storylines described in Section 2.1 were used as initial and boundary conditions to drive the ICON (ICOsahedral Non-hydrostatic) model Version 2.6.5.1 (Zängl et al., 2015), in a regional climate configuration known as ICON-CLM (Pham et al., 2021). We use the runtime environment "Starter Package for ICON-CLM Experiments" SPICE v.2.0 (Rockel and Geyer, 2022) to conduct all the simulations. For each storyline, we simulate the full period 2017-2022 described in Section 2.1. The downscaling is performed on the Euro-CORDEX domain (Jacob et al., 2014) at R12B5 resolution corresponding to a horizontal grid spacing of 0.11° or 12 km, hereafter referred to as EUR-12 (see Fig. 1a, see e.g., (Prill et al., 2023) for the description of icosahedral grid spacing conventions). Subsequently, we run a nested ICON-CLM simulation at R13B7 resolution, corresponding to 0.0275° or 3 km horizontal grid spacing on ensemble member 1 for the extended German domain GER-3, including the peripheral hydrological catchment areas (see Fig. 1b). In this study, we only focus on ensemble member 1 for the GER-3 domain. The detailed analysis of the full GER-3 ensemble is thus reserved for another study.

In the upper boundary, grid point nudging was applied in the ICON-CLM simulations to maintain proximity to the present-time circulation represented by the global spectrally nudged AWI-CM1 runs. In ICON, this nudging is implemented as an

additional forcing term that is being added to the prognostic equations at each fast physics time step (Prill et al., 2023):

$$\psi(t) = \psi^*(t) + \alpha_{nudge} N_{ds} [\psi_{bc}(t) - \psi^*(t)] \tag{1}$$

Where $\psi_{bc}(t)$ is the value of the prognostic variable $\psi$ at the time $t$ taken from the driving model, $\psi^*$ is the value of the variable $\psi$ before the nuging, while $\alpha_{nudge}$ refers to the nudging strength, and $N_{ds}$ is the number of dynamics substeps per fast physics step (Prill et al., 2023). Upper boundary nudging is applied in a sponge layer of the chosen thickness, where the nudging strength increases quadratically with height, starting with zero at the nudging start height $z_{start}$ and reaching the maximum nudging coefficient $B_0$ at the model's top height:

$$\alpha_{nudge} = B_0 \left( \frac{z - z_{start}}{z_{top} - z_{start}} \right)^2 \tag{2}$$

The nudging coefficient for the thermodynamic prognostic variables $\theta_v$, $\rho$, and $q_v$ was set to zero to prevent overfitting and, even more importantly in the context here, to allow for the free development of thermodynamics. Thus, upper boundary nudging was applied only to the horizontal velocity. We kept the maximum coefficient $B_0$ at its default value of 0.04. The nudging start height $z_{start}$ in the EUR-12 domain was set to 5000 m, while for the GER-3, it was left at 10500 m to prevent interaction with deep convection. Further information can be found in Prill et al. (2023).

Soil temperature and soil moisture data from ERA5 were used for soil initialisation in all EUR-12 simulations due to the partial unavailability of soil temperature in AWI-CM1 outputs. In ICON, the initialising soil data is pre-processed and remapped onto the 8-layer mesh by the built-in algorithm (Prill et al., 2023; Pham et al., 2021). To account for the possible discrepancy of soil types between ICON and ERA5, the volumetric soil moisture is transformed into the universal soil moisture index (SMI), which makes it independent of the soil type (Prill et al., 2023). For the present-time experiment, the initial simulation year (2017) was considered for spin-up. In the storyline simulations, an additional year was required for the soil to adapt to the warmer climate. Therefore, we ran the dynamical year 2017 twice. Additionally, the temperature of the bottom soil layer, which is not prognostic in the TERRA land module of ICON but is set to the climatological annual mean near-surface temperature T_CL based on the Climate Research Unit data (Schulz et al., 2016; Mitchell and Jones, 2005), was adjusted to reflect the respective global warming level for each storyline. The lower boundary condition for soil moisture is given by a free-drainage formulation and thus did not require additional adjustments (Prill et al., 2023; Chen et al., 2018; Zeng and Decker, 2009).

### 2.3 Model evaluation approach

We compare the obtained daily 2m temperature fields to the ERA5 reanalyses (Hersbach et al., 2020), as well as to the daily gridded land-only observational dataset over Europe E-OBS v.28 (see e.g., Cornes et al., 2018). To evaluate the simulations on the station level, we use 20 stations of the German National Weather Service (DWD, 2023; Kaspar et al., 2013, see the full list in Table S1). The root mean square difference (RMSD) to observational datasets (DWD and E-OBS) and its change between simulations of different resolutions ($\Delta$RMSD) is chosen as a metric to compare the representation of near-surface temperature by the models of our GCM-RCM-CPM chain in the present-day storyline. The significance of $\Delta$RMSD between ICON EUR-12 and GER-3 is computed with the paired difference test (Rubin, 1973).

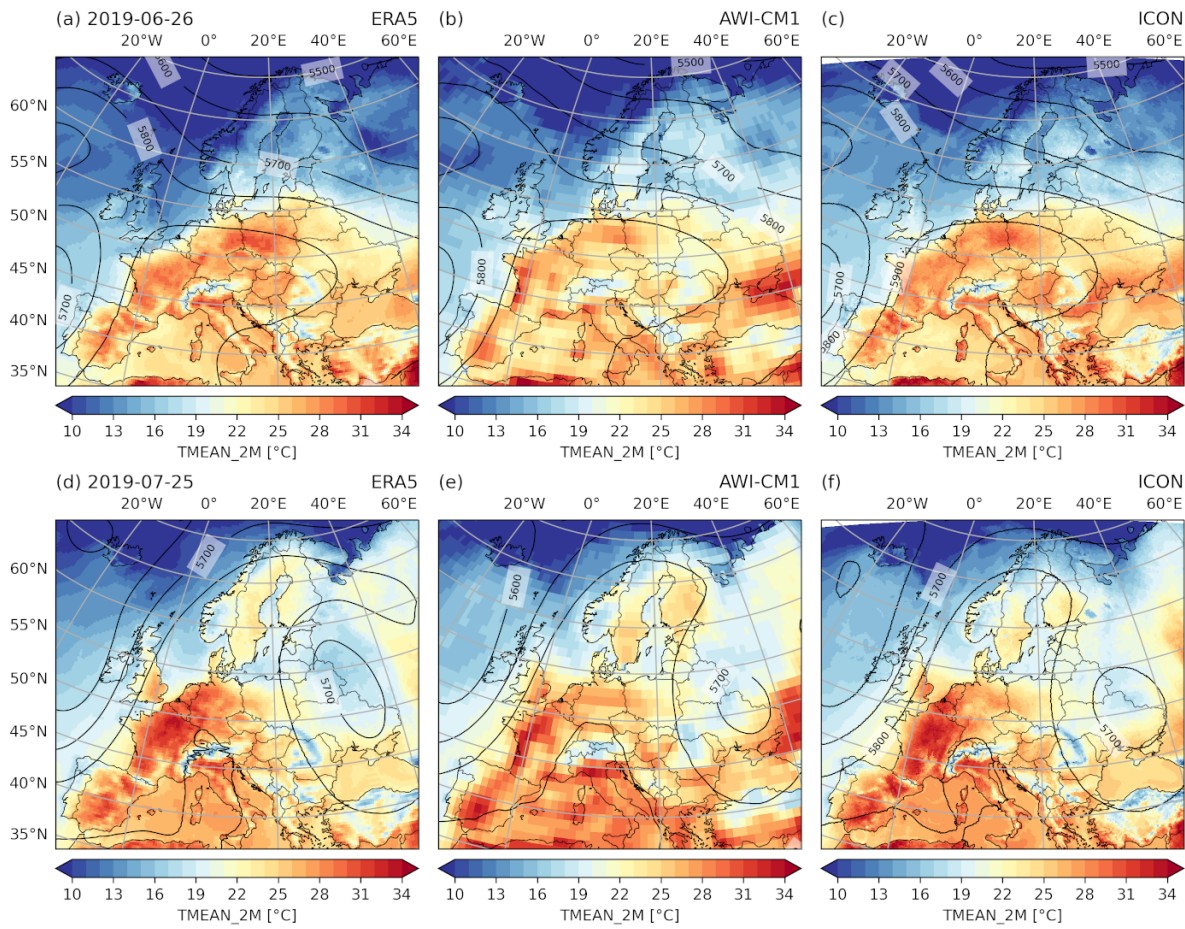

**Figure 2.** Mean 2m temperature (shading) and geopotential height at 500 hPa (contours) on the 26th of June 2019 (the first peak of the June heatwave) and 25th of July 2019 (the peak of the July heatwave) for ERA5 (left), AWI-CM1 (middle), and ICON EUR-12 (right).

## 3 Results

### 3.1 Evaluation of the present-day storylines

The results of the ICON regional model simulations are evaluated by comparing the model output with observational E-OBS and ERA5-reanalysis data. The comparison of the 2m temperature and 500-hPa geopotential height fields for the heatwave peaks of June and July 2019 shows a good agreement between the ICON EUR-12 simulations (Figure 2c,f) and ERA5 reanalyses (Fig. 2a,d). Furthermore, the ICON simulation shows a clear improvement in the representation of daily 2m temperature compared to the global AWI-CM1 simulation (Fig. 2b,e). The comparison of the daily temperature fields to E-OBS yielded similar results (see Fig. S1)

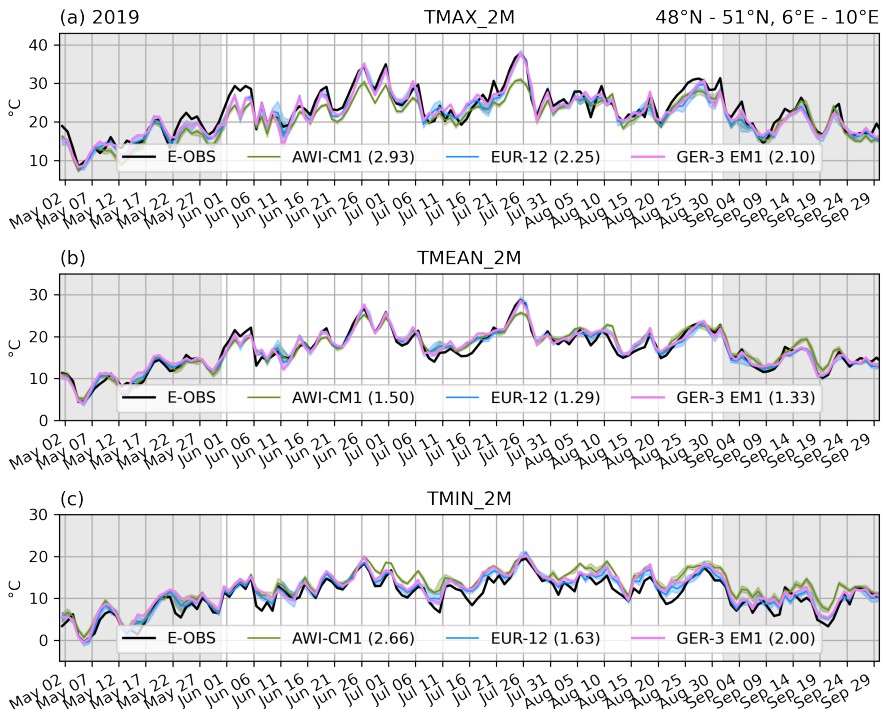

**Figure 3.** Comparison to E-OBS of (a) daily maximum, (b) daily mean, and (c) daily minimum 2m temperatures obtained with AWI-CM1, ICON EUR-12, and ICON GER-3 (EM1) averaged across the longitude-latitude box with the boundaries 48° N - 51° N, 6° E - 10° E (dashed box in Fig. 1b). Shading spans the minimum/maximum range of values obtained from the 5-member ensembles. The numbers in brackets in the legend report the RMSD of ensemble member 1 to E-OBS in June,July, and August.

In Fig. 3, the time series of daily maximum, mean, and minimum 2m temperatures are compared to E-OBS over the longitude-latitude area 48° N - 51° N, 6° E - 10° E, to investigate the underestimation of maximum temperature during the exceptionally hot periods in June and July 2019 mentioned in Sánchez-Benítez et al. (2022). The ICON simulations show an improvement both for the daily maximum and minimum temperatures in July in this area. This indicates a more accurate representation of the diurnal temperature range in the regional ICON simulations.

We compared the performance of the simulations within the model chain by calculating the root mean square difference (RMSD) in June - August of the simulated 2m temperature with respect to DWD observations (DWD, 2023) using 20 selected stations (locations are shown in Fig. 1b; the full list of stations is given in Table S1 of the supplementary materials). The RMSD is significantly reduced by the dynamical downscaling of AWI-CM1 data to the EUR-12 domain (see Table 1). In the case of the GER-3 simulation, a further reduction of the RMSD could be achieved only for the daily maximum 2m temperatures, whereas daily minimum and mean temperatures slightly deteriorated but remained clearly improved relative to AWI-CM1. A similar result can be obtained when the RMSD is calculated between the simulations and E-OBS time series shown in Fig. 3 (RMSD values in the legend).

**Table 1.** The root mean square differences (RMSD) in °C of daily maximum, mean, and minimum 2m temperature to DWD station observations during summer (June to August) 2019 averaged over 20 stations ( for locations, see Fig. 1)

|  | AWI-CM1 | ICON EUR-12 | ICON GER-3 |
|---|---|---|---|
| TMAX_2M | 3.99 ($\sigma$=1.35) | 2.88 ($\sigma$=0.62) | 2.79 ($\sigma$=0.46) |
| TMEAN_2M | 2.20 ($\sigma$=1.24) | 1.66 ($\sigma$=0.45) | 1.73 ($\sigma$=0.50) |
| TMIN_2M | 3.21 ($\sigma$=1.08) | 2.37 ($\sigma$=0.62) | 2.67 ($\sigma$=0.43) |

Figure 4a-c shows the spatial distribution of RMSD for EUR-12 with respect to E-OBS for daily maximum, mean, and minimum temperatures during the summer of 2019. The RMSD varies between 1 and 2 °C in Central Europe for the daily mean temperature and between 2 and 3.5 °C for the daily minimum and maximum temperatures, which supports the values shown in Table 1. We interpolated the ICON EUR-12 data to the grid of AWI-CM1 and compared the RMSD of both models to E-OBS. Green colours in Fig. 4d-f indicate an improvement in the performance of the regional simulations compared to the global AWI-CM1 simulation for the respective temperatures. The overall improvement in central and southern Europe is robust, with the exception of the maximum temperature at the coast of the Iberian Peninsula and north of the Black Sea.

To identify potential systematic biases, the observed E-OBS temperature fields were compared to the EUR-12 ones. From Figures S2 and S3 it can be inferred that the negative bias of daily maximum temperatures in central Europe introduced in AWI-CM1 simulations is reduced (see Fig. S3a,d) along with the positive bias of daily minimum temperatures(Figure S3c,f). According to Fig. S3a,g, the increased error of daily maximum temperatures in western Iberia and Eastern Europe mentioned above can be attributed to the positive bias introduced by ICON-CLM.

A similar comparison was conducted for all simulated summer seasons between 2018 and 2022 (Fig. S and S5). It demonstrates that the bias patterns persist consistently across all the simulated summers, suggesting that the errors are not flow-dependent.

The nested convection-permitting GER-3 simulation was assessed by comparing the RMSD of the 2m temperature to observations with the RMSD of the driving EUR-12 simulation (see Fig. 5). With this aim, both temperature datasets were interpolated to the E-OBS grid (0.1° horizontal resolution). We also evaluated the average bias between the nested simulations by subtracting the EUR-12 seasonal mean temperature fields from the GER-3 fields coarsened to the EUR-12 domain (see Fig. S6a-c), as well as the bias of the GER-3 fields with respect to E-OBS (Fig. S6d-f).

The nested GER-3 simulation is between 0.5 and 2 K warmer than the EUR-12, which further reduces the negative bias of the daily maximum temperature over Germany. However, the positive bias of the daily minimum temperature increases and the daily mean temperature representation shows no robust improvement over Germany. The most substantial bias of daily maximum temperature occurred in the most western parts of the domain over land and in the Po Valley. However, those areas are located inside of the lateral boundary zone and are not used in the analyses (see Fig. S6). Due to refined topography, the nesting significantly reduced the RMSD of maximum and mean 2m temperature over the Alps (see Fig. 5d-f).

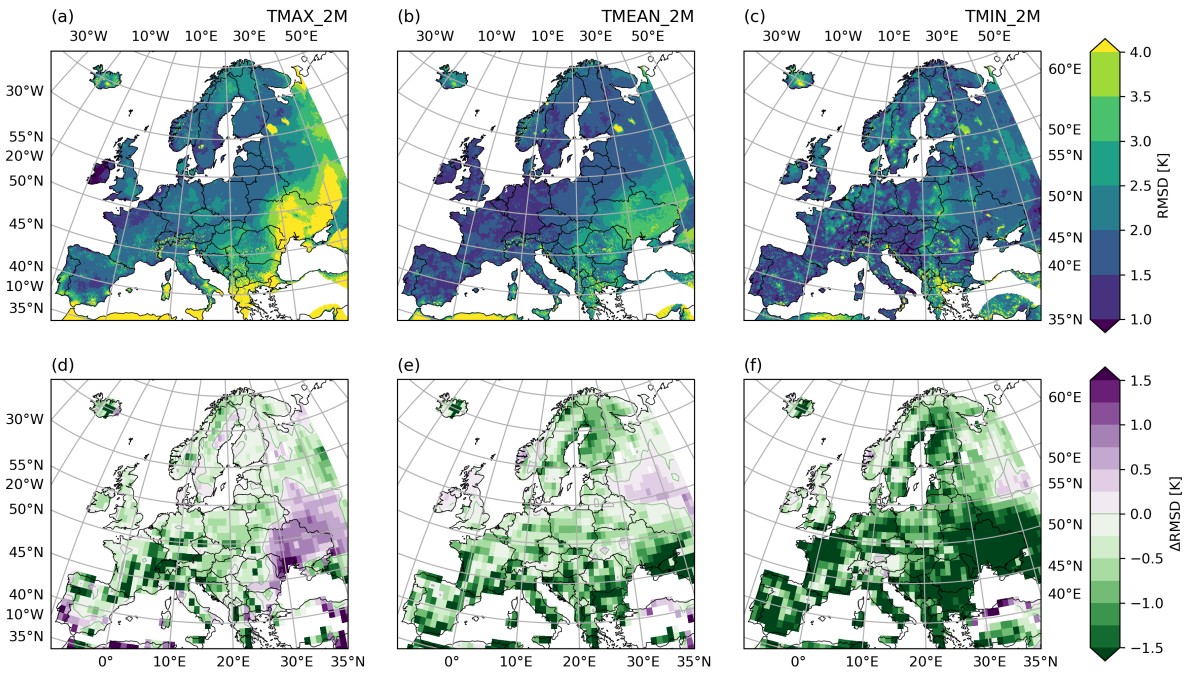

**Figure 4.** Performance assessment of the ICON EUR-12 simulation for June, July, August 2019 for daily maximum (left column), mean (middle column), and minimum (right column) 2m temperature. (a-c): root mean square difference (RMSD) of the simulated daily 2m temperatures by ICON EUR-12 to E-OBS. (d-f): Change in RMSD achieved by dynamical downscaling; the green colours correspond to the reduced squared error of daily temperatures.

Analogously to the EUR-12 simulation, the evaluation of GER-3 simulation over the full simulation period yields that the patterns shown in Fig.5 remain consistent over the years for all modelled summer seasons between 2018 and 2022 (see Fig. S7 and S8 in supplementary materials), with a more prominent improvement of RMSD for the maximum temperatures (Fig. 5d and S7d).

## 3.2 Storylines for the summer 2019 heatwaves

Given the improved performance by dynamical downscaling with ICON-CLM for present-day conditions, we now analyse the regionalised past and future analogues of the July 2019 European heatwave.

We first consider the period corresponding to the peak of the July 2019 heatwave. According to Fig. 6, the maximum temperature on the 25th of July would exceed 40 °C over a considerable area of Western Europe for a +4 K climate. The area affected by temperatures exceeding 40 °C is projected to increase significantly in the EUR-12 simulations, from $21{,}000 \pm 14{,}000 \, \mathrm{km}^2$ in the pre-industrial climate to $290{,}000 \pm 40{,}000 \, \mathrm{km}^2$ in the present-day and $1{,}110{,}000 \pm 70{,}000 \, \mathrm{km}^2$ in the +4 K climate (see

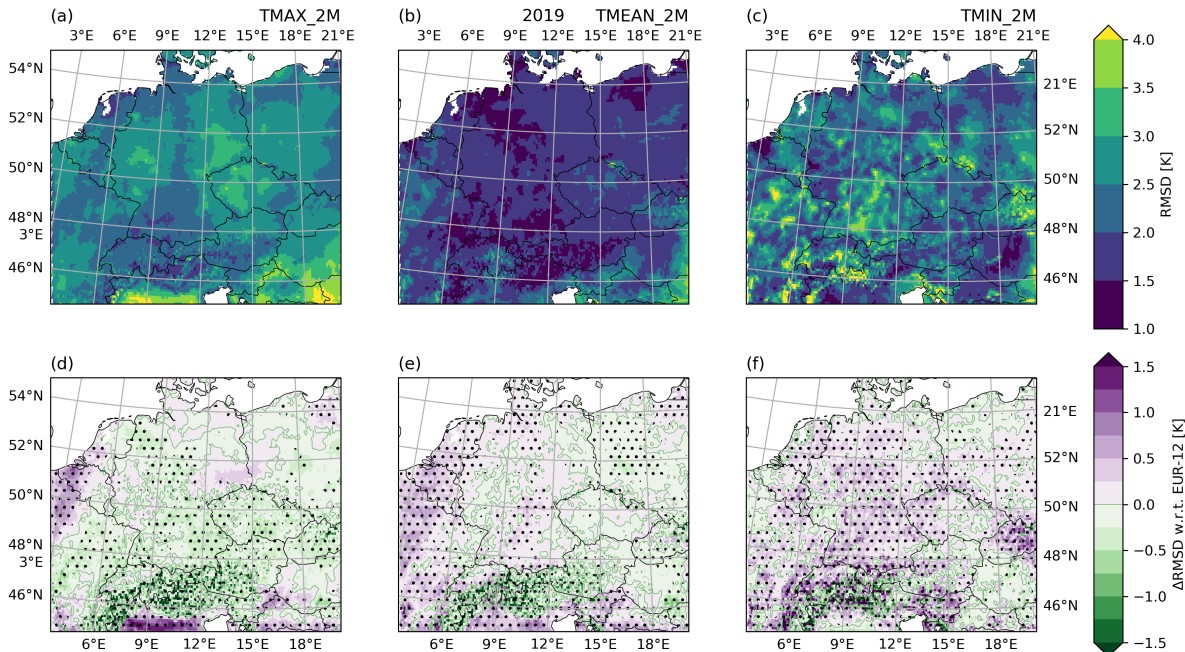

**Figure 5.** The evaluation of the nested convection-permitting GER-3 simulation for June, July, August 2019 for daily maximum (left column), mean (center column), and minimum (right column) 2m temperature. (a-c): root mean square difference (RMSD) of daily 2m temperature to E-OBS of the GER-3 simulation. (d-f): Change of RMSD compared to the EUR-12 simulation (significant difference hatched, p<0.05).

Fig. S9). Moreover, the 45 °C threshold would be exceeded over a large area in Western France already in a +3 K climate, and
for the Benelux and Rhine valley in a +4 K warmer world (Fig. 6i and j).

The increase of the 2m temperature is not spatially homogeneous, with the regions located outside of the heatwave's core experiencing stronger warming, thereby contributing to the increasing spatial extent of the heat wave. This is exemplified in Fig. S12, where the maximum temperature differences between the +4 K and pre-industrial climates reach 12 °C in Luxembourg, Southern Belgium, Western Germany, and the most eastern parts of France. The temperature increase in the core of the
heatwave is close to 8 °C, which corresponds to a doubling of the global warming level (+4 K).

Figure 7 displays the time series of daily maximum, mean, and minimum temperatures averaged over a longitude-latitude area 48° N - 51° N and 6° E - 10° E (depicted in Fig. 1b) for the extended summer season (May to September) of 2019. During July and August, an increase in spread is found between the temperature curves corresponding to the different warming levels. Conversely, the spread between the time series of temperatures is much smaller in May and early June. This finding was
previously confirmed in the analyses of the global storylines by Sánchez-Benítez et al. (2022) and will be further explored in the following section. We also see a larger spread in daily maximum curves in July and August compared to the mean and minimum temperatures.

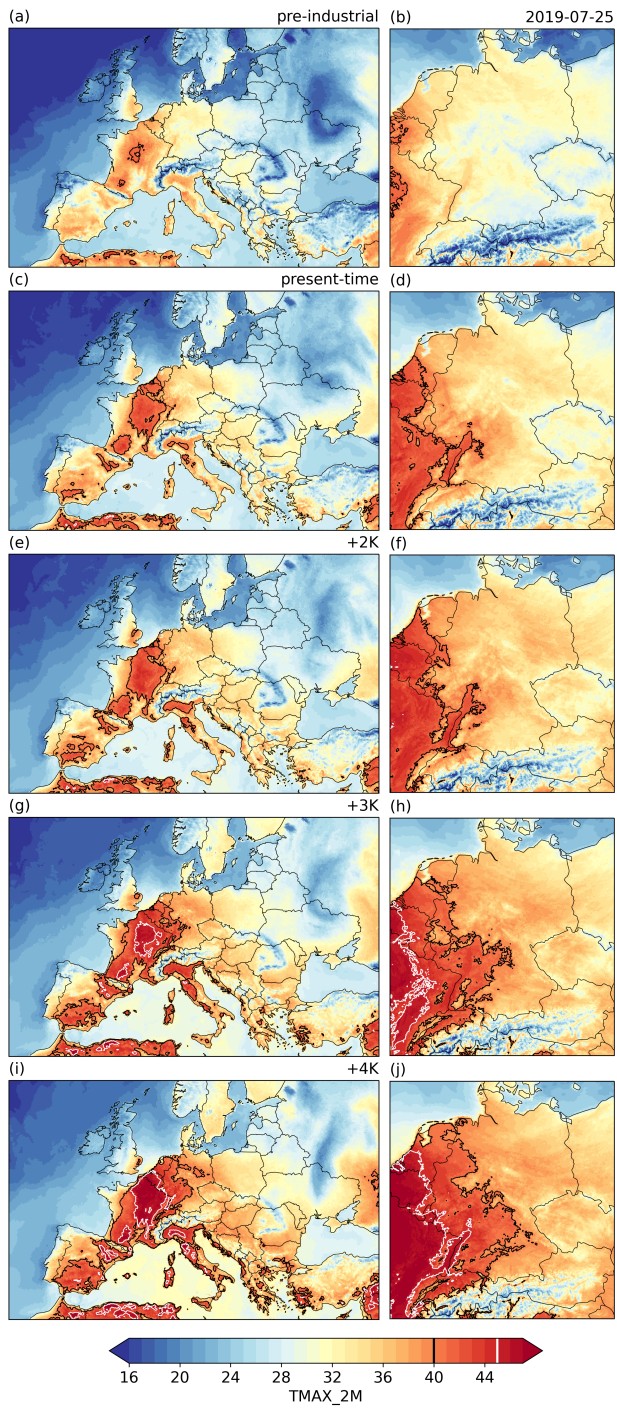

**Figure 6.** Daily maximum 2m temperature on the 25th of July as of the (left column) ICON EUR-12 simulations and (right column) ICON GER-3 simulations in pre-industrial (a, b), present-time (c, d), +2 K (e, f), +3 K (g, h), and +4 K (i, j) climates. Based on the ensemble member 1; for the ensemble spread of EUR-12 simulations, see Fig. S11.

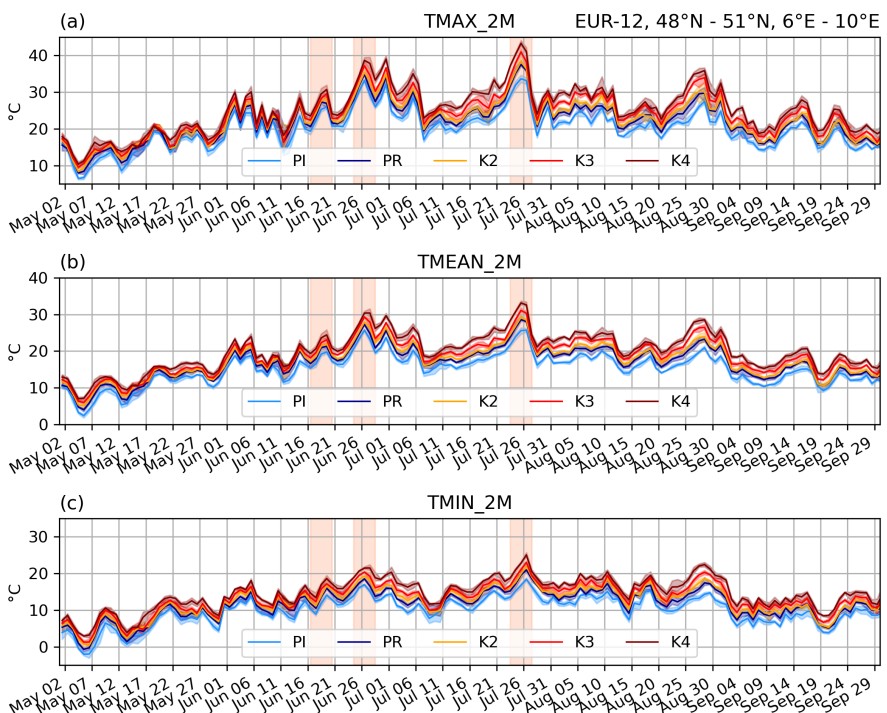

**Figure 7.** Daily (a) maximum, (b) mean, and (c) minimum temperatures averaged over the longitude-latitude box with boundaries 48° N - 51° N and 6° E - 10° E (see Fig. 1b) over the MJJAS period of the year 2019 based on the EUR-12 storyline simulations. Shading spans the minimum/maximum range of values obtained from the 5-member ensembles. The three highlighted periods (orange) are discussed in detail in section 3.3. For the GER-3 simulations, see Fig. S13.

## 3.3 Temperature scaling in response to global warming

In order to gain a deeper insight into the spread of temperature curves depicted in Fig. 7 and to address the question of temperature scaling in response to global warming, three five-day periods were selected for detailed analysis (highlighted orange in Fig. 7). The first period is in mid-June, when no heatwave was observed, while the second period is in late June during the first heatwave, and the third period is around the peak of the July heatwave. The daily maximum, mean, and minimum 2m temperatures averaged spatially over the area 48° N - 51° N, 6° E - 10° E and temporally over those periods show a clear linear dependency with the global warming level (see Fig. 8a-c). Therefore, we express anthropogenic change to the 2m temperatures per 1 °C of global warming as a slope of this line. This slope will be referred to in the following text as the "warming rate."

As shown in Fig. 8a, the average warming rate over the studied area is close to a factor of 1 in mid-June for all three curves. This indicates that the warming rate is comparable to global warming in the absence of an extreme event in early summer. The warming rates increase during the first heatwave and approach or even exceed a factor of 2 for maximum temperature during

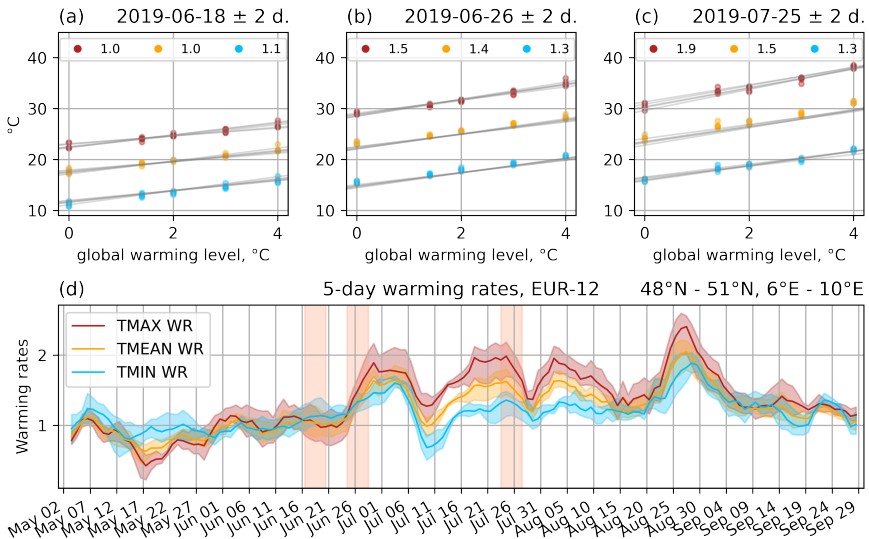

**Figure 8.** (a-c) Daily maximum (red), mean (orange), and minimum (blue) 2m temperature over the longitude-latitude box 48° N - 51° N, 6° E - 10° E averaged over three 5-day periods plotted against the global warming level. The numbers in the legend represent the ensemble means of the slopes of the respective lines; (d) warming rates for the rolling average (5-day window) of daily maximum, mean, and minimum temperatures over the same box. Shading spans the minimum/maximum range of values obtained from the 5-member ensembles. The three highlighted periods are discussed in detail in section 3.3. Based on the EUR-12 simulation. For the GER-3 simulation, see Fig. S14.

the July heatwave (Fig. 8b,c). This suggests that the expected differences for the maximum temperature for a similar event in a future climate would be exacerbated and be twice as high as the corresponding global warming level.

Using this method, we computed the warming rates for 5-day running mean maximum, mean, and minimum temperatures (see Fig. 8d) for the extended summer season of 2019. Between May and early June, the warming rates fluctuate around 1. The values increase with the onset of the first heatwave in late June, with the maximum temperature responding stronger than the minimum temperature. Following a relative minimum in mid-July, the warming rates strongly increase before the 25th of July, reaching a factor of 2.2 for maximum temperature ahead and during the late July heat wave. After a short decrease directly after the temperature maximum, the warming rates increase again to values over 2.0 at the beginning of August. This is followed by a gradual decrease before the last peak of warming rates with even higher magnitude, which occurs in late August when the temperature again increases during the summer. We see a general tendency for the warming rates to rise towards late summer, which is possibly associated with the gradual decrease in soil moisture. The increase of global warming amplification in Central Europe has also been obtained by Sánchez-Benítez et al., 2022 in both free and nudged AWI-CM1 simulations.

To comment on the broader warming rate peak ahead of the July heatwave that can be seen in Fig. 8d, we estimate the duration of this heatwave in different climates based on the exceedance of the 90th percentile of the modelled maximum 2m temperature, which is 30 °C when computed for all days in July over the simulated period of 2018-2022. According to our

estimation, the duration of this heatwave would grow highly non-linearly from 4 days in the present-day climate to 5 days in the +3 K climate, and to 9 days in the +4 K warmer world (not shown).

While the response of the maximum temperature in May, June, and the second half of September is smaller or comparable to the response of the mean and minimum temperatures, the warming rate for the maximum temperature is very large during July and August. Thus, with global warming, the diurnal temperature range tends to increase in the mid and late summer of the dynamical year 2019. In contrast, the temperature response appears to be distributed more uniformly during the day in spring, early summer, and early autumn. We obtained similar behaviour in other summer seasons (May - September) of the simulated period.

The warming rates are now computed for each grid point in the study area. Figure 9a displays the warming rates during the five-day period around the peak of the heatwave in late July 2019 (the third shaded area in Fig. 7) at each grid point in both the EUR-12 and GER-3 domains. According to the goodness of fit maps in Fig. S15, the assumption of linear growth in the areas affected by the heatwave is valid in all cases. For comparability with the GER-3 simulations, Figure 9 shows warming rates obtained from the ensemble member 1 of both domains. The ensemble mean of the temperature response in the EUR-12 simulations is shown in Fig. S16 of the supplementary materials, which depicts similar patterns with small differences in detail.

The warming rates for the daily maximum temperatures exceed a factor of 2 over large areas in central and southern Europe (see Fig. 9a). In contrast, the minimum temperatures increase at a comparatively lower rate. This also indicates an enhanced diurnal temperature range during extreme heat events, as discussed above. Consistent with the values shown in Fig. 8a, the warming rates during a "neutral" period in early summer are much lower and below a factor of 1 across Central Europe (Fig. S17a).

A closer look into the response of the 2m temperature on the 25th of July reveals that the warming rates during the peak of the event reach a factor of 3.0 east of the heatwave's core, underscoring that those areas would become up to 12 °C warmer in the +4 K climate compared to the pre-industrial time (see Fig. 9b). The mean warming rates of the EUR-12 ensemble show a similar pattern: the black contour in Fig. S16 encompassing the core of the heatwave in present-day climate does not coincide with the 2.5 K/K contour of the temperature response. Accordingly, the spatial extent of the heatwave is subject to a rapid increase in future climates. As shown in Fig. S9, the area affected by the maximum temperatures over 40 °C on the 25th of July grows with the global warming level at an approximate rate of $270,000 \pm 1,300 \ \text{km}^2\text{K}^{-1}$, slightly accelerating towards +4K climate. The lack of warming over the British Isles may be associated with the fact that this area is surrounded by sea and/or located at the edge of the subtropical ridge triggering this event and, thus, less affected.

To compare the scaling of the temperatures during the extreme event to the mean scaling, we estimated the average monthly response of the 2m temperatures over Europe for five simulated summers from 2018 to 2022. According to Fig. 10, the warming rates are close to 1 in Central Europe in June, increase to 1.5 in July, and approach 2.0 in August. Such an intensification of the temperature response towards late summer indicates a higher risk of heatwave development in the warmer world. This aspect was investigated in Hundhausen et al. (2023), where, based on the ensemble of regional high-resolution climate simulations, the probability of large heatwaves was found to gradually increase during the summer.

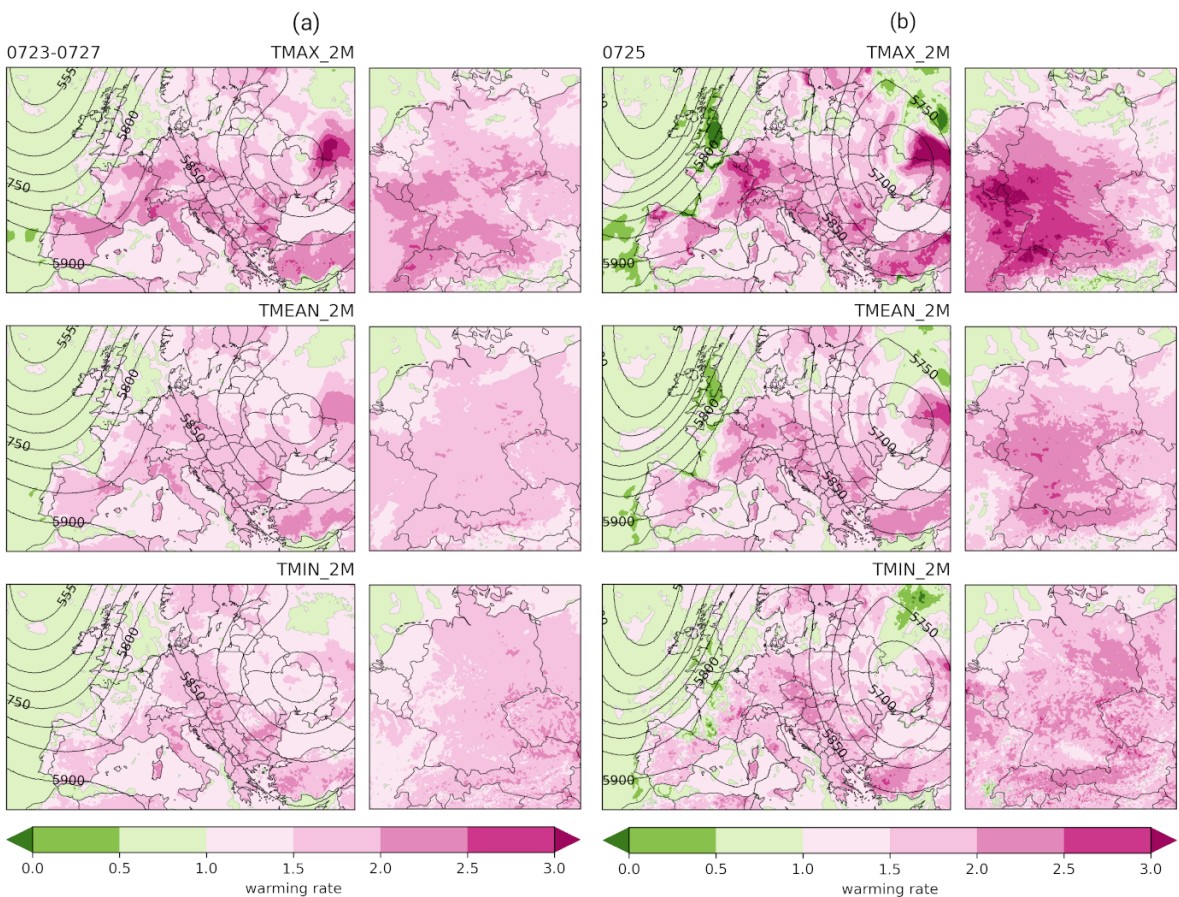

**Figure 9.** (a) Warming rates for the period from the 23rd to the 27th of July 2019. (b) Warming rates for the 25th of July. Contours: geopotential height as of the EUR-12 simulation. The figure is based on the ensemble member 1.

Comparing the mean scaling in Fig. 10 to the warming rates in Fig. 9b, we find that the scaling of the maximum temperature
at the peak of the July 2019 heatwave is nearly twice as high in Western France and Eastern Germany. Nevertheless, it is worth
mentioning that all the years of the simulated period lay within the European multi-year drought and heat event of 2018-2022
(Knutzen et al., 2023). Thus, the average warming rates in Fig. 10 may be different from those for the years unaffected by
severe drought conditions.

## 4   Summary and Discussion

In this study, we follow an event-based storyline approach using a GCM-RCM-CPM model chain to analyse the thermodynamic
response of the European summer 2019 heatwaves to global warming. We obtained our storylines using spectrally nudged

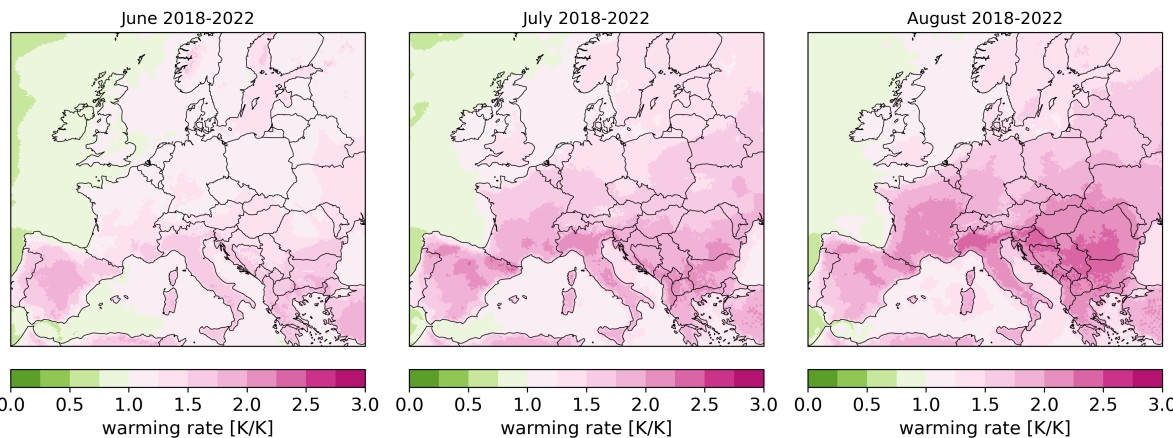

**Figure 10.** Mean warming rates for the maximum 2m temperature in June, July, and August 2018-2022 based on the EUR-12 storylines.

global AWI-CM1 simulation and subsequent dynamical downscaling with the regional model ICON-CLM to resolutions of 12 km (EUR-12) and 3 km (GER-3).

The outcomes of the simulations permit the answering of the three key research questions:

(1) How accurately can a regional event-based storyline simulation represent a recent event, and what is the improvement compared to the global spectrally nudged storyline simulation?

The daily spatial and temporal patterns of the 2m temperature fields obtained with the AWI-CM1 - ICON model chain show good agreement with ERA5, E-OBS, and DWD station observations for the summer of 2019. Compared to the driving AWI-CM1 simulations, the dynamical downscaling significantly reduced the RMSD of 2m temperature over most of Europe

(by about 1.5 °C in Central Europe). The GER-3 simulation gives more spatial details to the EUR-12 for the daily maximum temperature; however, it does not necessarily improve the representation of daily minimum temperatures.

(2) What is the effect of climate change on the 2019 European heatwave based on the regional and convection-permitting ICON-CLM simulations?

Based on the simulations for our case study in 2019 with different thresholds for global warming, the peak temperatures

of the July heatwave would increase considerably beyond the underlying global warming level, with the magnitude of the temperature response depending on the location. The increments of daily maximum temperature in the +4 K climate with respect to the pre-industrial climate vary between 8 °C in the centre of the heatwave and 12 °C to the east of it. This leads to the increased spatial extent of the heatwave in the warmer world. In the context of the present-day climate, where we took the 40 °C isoline as a benchmark, the affected area would be encompassed by the 45 °C isoline in a +4 K world. Consequently, the areas

affected by temperatures exceeding 40 °C would experience a significant expansion, exceeding 1,000,000 km$^2$ in the +4 K storyline. When considering the time series of 2m temperature over the area 48° N - 51° N, 6° E - 10° E for all five storylines, the spread between the curves appeared to be higher in mid and late summer compared to the early summer. This aligns with

the previous findings that suggest intra-seasonal dependency (intensification towards late summer, see e.g., Hundhausen et al., 2023) of anthropogenic warming based not only on the nudged storylines but also on the free CMIP6 runs (Sánchez-Benítez et al., 2022).

(3) What is the local to regional extreme temperature scaling in response to global warming for an event like the 2019 heat wave?

Our findings reveal a linear dependency of the 2m temperature response to the global warming level, with the derived warming rate determined by the slope of the linear regression demonstrating spatial and intra-seasonal variations. Quantifying the smaller spacing between the temperature curves mentioned above, the warming rates over the studied area in the early summer fluctuate around 1, indicating that regional warming aligns with global warming for that period. However, in July and August, the warming rates for daily maximum temperatures vary between 1.5 and 2.5, reaching higher values with each successive heatwave during the study period (see Fig. 8d).

Furthermore, the broadening of the warming rate peak during the July heatwave means an extension of the heatwave's duration in a warmer world. We also observe the broadening of the diurnal temperature range in future climates, which is indicated by much lower warming rates for the minimum temperature than those for the daily maximum. This difference does not occur in early summer and disappears again by late September. Considering that the first heatwave in late June had modified the soil moisture for the rest of the summer (Sousa et al., 2020; Sánchez-Benítez et al., 2022), this case exemplifies the dependency of the global warming amplification on the event-specific regional evolution of the thermodynamic conditions.

On the 25th of July, the response of the maximum 2m temperature (warming rate) reached a factor of 3 in some areas (Fig. 9b). However, the highest warming rates are not located over the heatwave centre but instead shifted eastward (see Fig. S12 and S16). The areas less affected by unprecedented temperatures but still located within the influence of the event triggered by the subtropical ridge tend to heat stronger in the warmer world. Thus, along with the increasing duration, the area affected by the heatwave is expanding. While the GER-3 domain effectively captures the regions with the strongest temperature responses, it does not fully encompass the core area affected by the July heatwave. We admit that for a more comprehensive km-scale investigation of this heatwave, this domain should have covered a larger fraction of France. Still, as this study is part of the Innopool SCENIC project (Helmholtz Changing Earth, 2024), which focuses on the impacts of climate change within Germany, our CPM simulations provide the necessary insights for this context.

Several factors may have influenced the spatially variable magnitude of the 2m temperature response, which is seen both for a single member (Fig. 9) as well as for all ensemble members (Fig. S16). One possible explanation for this behaviour is a heterogeneous response of soil moisture-temperature coupling within and outside the heatwave's core (see e.g., Gevaert et al., 2018; Miralles et al., 2014). We hypothesise that the overall amplification of the warming rates during heatwave events and the extension of the diurnal temperature range may have been exacerbated due to soil-atmosphere feedback. However, a dedicated and detailed analysis would be necessary to demonstrate or dismiss this hypothesis, as other factors like small changes in atmospheric dynamics may have also played an important role.

The same event was analyzed by Vries et al., 2024 using a PGW approach, yielding similar temperature responses of 1.5 to 2.5 K/K during heatwaves. Both studies show a higher response of maximum temperatures compared to that of minimum

temperatures. However, unlike our findings, they observed no significant response dampening within the heatwave core. Nevertheless, both studies indicate that areas in France impacted by extreme temperatures on the 25th of July do not show higher scaling than the surrounding areas. While Vries et al., 2024 focuses on the southeastern Netherlands, where higher temperatures yield stronger responses (see Fig. S10 in Vries et al., 2024), this relationship may not be directly applicable to Central France for events as extreme as the July 2019 heatwave, given the region's distinct climate, and further investigation is needed to determine whether such scaling applies. The complementary results from both studies enrich our understanding of heatwave dynamics and provide a broader context for future investigations.

## 5  Conclusions

The aim of this study was to provide a regional perspective of the global spectrally nudged storylines for the summer heatwaves of 2019 in Central Europe. We addressed the unfolding of the heatwaves on the regional-to-local spatial scales and followed the evolution of the near-surface temperatures throughout the whole summer season in five dynamical analogues of the summer of 2019 by a dynamical downscaling approach. We observed that the late June heatwave triggered higher warming rates and an extension of the diurnal temperature range in a warmer world for the rest of the summer. Additionally, we obtained the higher warming rates over the regions east of the July heatwave centre, as well as the broadening of the warming rate peaks associated with both 2019 heatwaves. This demonstrates that our approach allows not only for the estimation of possible impacts of extreme heat events in the warmer world but also for the investigation of the mechanisms and conditions that lead to different rates of response to background warming.

Our regional storylines can be used to drive hydrology, land surface, and other impact models that will deliver relevant information for developing adaptation measures. Finally, the insights gained from storyline-based regional impact studies are more tangible than probabilistic estimates and, thus, bear the potential to raise public awareness about the significance of the effects of climate change on the community level.

*Code and data availability.* The ICON model is available as open source release since January 2024 under https://icon-model.org/. The runtime environment SPICE v.2.0 is available online under https://zenodo.org/records/6838984. The simulations are stored on the supercomputer Levante at the German Climate Computation Center (DKRZ, Hamburg) and will be made available online upon completion of the data preparation. ERA5 data can be downloaded from the Copernicus Climate Change Service (C3S) Climate Date Store (https://cds.climate.copernicus.eu) and can be accessed at the DKRZ by the users of the Levante HPC system. The E-OBS dataset can be accessed at the website of the European Climate Assessment & Dataset project (https://www.ecad.eu/download/ensembles/download.php). The DWD station data are freely available for research at the Open Data Portal of the German Weather Service DWD (https://opendata.dwd.de, DWD, 2023).

*Author contributions.* TK, PL and JGP conceived and designed the study. ASB computed the global AWI-CM1 simulations and provided the data and necessary instructions. PL took care of the pre-processing of the input data to ICON. TK performed the ICON simulations with the help of PL, performed the data analysis and prepared the figures. HG and PB contributed additional modelling expertise. TK wrote the

initial paper draft. All authors discussed the results and contributed with manuscript revisions.

*Competing interests.* The contact author has declared that none of the authors has any competing interests.

*Acknowledgements.* This work was supported by funding from the Helmholtz Research Field Earth & Environment for the Innovation Pool Project SCENIC. JGP thanks the AXA research fund for support. The authors thank the German Climate Computation Center (DKRZ, Hamburg) for providing computing and storage resources under projects 105 and 1264. The global AWI-CM1 simulations were performed

using ESM Tools (Barbi et al., 2021). The authors further thank Klaus Keuler for providing the ICON-CLM settings and domain grids used within the NUKLEUS project and Florian Ehmele for establishing the workflow of the dynamical downscaling and producing the set of preliminary simulations.

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
