# Peer review of "The European summer heatwave 2019 - a regional storyline perspective"

_Earth System Dynamics, 2024_

## Referee Comment (RC2)

Journal: Earth System Dynamics, discussion paper esd-2024-16
Review, 27 August 2024

Title: "The European summer heatwave 2019 – a regional storyline perspective"

Authors: Tatiana Klimiuk, Patrick Ludwig, Antonio Sanchez-Benitez, Helge F. Goessling, Peter Braesicke, and Joaquim G. Pinto

Recommendation: [Major/Minor Revision]

**GENERAL COMMENTS**:

This article presents a storyline approach of the unfolding of European summer of 2019 on a regional scale with special focus on the heat wave end of July 2019. Next to assessing the outcomes for present-day conditions, corresponding to +1.4K global warming, the methodology is repeated to cooler pre-industrial conditions, and to warmer future climates projected at several levels of future global warming. Storylines are constructed from a GCM-RCM-CPM model chain where the GCM is spectrally nudged to ERA5 vorticity and divergence to enforce the GCM atmospheric circulation to stay close to observed circulation patterns. GCM states are subsequently used for downscaling at 12 km for the European domain and 3 km for a Central-European region encompassing Germany. A performance assessment of the present-day climate simulation indicates an improvement of representing 2-meter temperature by the RCM and even more so by the CPM compared to the GCM. A primary finding of the storyline approach is a doubling, and at some locations, almost a tripling of local warming rates relative to the background warming during the episode prior and during the heat wave whereas earlier in the season (late spring/early summer) this ratio tend to be much smaller.

The storyline perspective built from constraining the climate model state through spectral nudging to a quasi-observed state (followed by standard dynamical downscaling) provides an elegant method to isolate the thermodynamic response to anthropogenically induced climate change from the circulation response. That part of the work is already developed and described in the paper by Sanchez-Benitez et al. (2022).
The present article focuses on the subsequent downscaling steps and potential impact on a regional to local scale. Overall it is well written and of general interest, however a number of issues require attention before the manuscript is suitable for publication.

**MAJOR POINTS**:

1. What I found quite surprising to notice is that the authors seem not aware, at least nowhere in the manuscript any reference is made, of a widely used alternative approach, commonly referred to as Pseudo Global Warming (PGW), in which following a comparable methodology storylines are built from primarily the thermodynamic (or physical) responses to projected future global warming. Examples in literature include for example:
   - Schär et al., 1996 Surrogate climate-change scenarios for regional climate models Geophys. Res. Lett. 23 669–72

- Prein et al., 2017.: The future intensification of hourly precipitation extremes, Nat. Clim. Change, 7, 48–52, https://doi.org/10.1038/nclimate3168
- Aalbers et al., 2023 The 2018 West-Central European drought projected in a warmer climate: how much drier can it get? Nat. Hazards Earth Syst. Sci. 23 1921–46 https://doi.org/10.5194/nhess-23-1921-2023
- Brogli et al., 2023 The pseudo-global-warming (PGW) approach: methodology, software package PGW4ERA5 v1.1, validation and sensitivity analyses Geosci. Model Dev. 16 907–26 https://doi.org/10.5194/gmd-16-907-2023

Moreover, just two weeks after this manuscript was submitted a paper by H. de Vries et al., entitled "Western Europe's extreme July 2019 heatwave in a warmer world", appeared in Environmental Research: Climate ( https://doi.org/10.1088/2752-5295/ad519f ) in which the authors develop a storyline perspective built on the PGW-approach focusing on the very same event.

In my opinion, reference to the PGW-approach in the context of the application presented in this paper is required, and I would strongly encourage to include a discussion on the pro's and cons of applying the respective methods (spectral nudging and PGW) in this type of storyline development, focusing on events or episodes.

2. It is unfortunate that the region with highest temperatures during the heat wave episode is on the western edge of the ICON-GER-3 domain (see Figure 6). In particular, it makes the conclusion that the region of highest relative warming rates shifts to the east questionable. Comparing both panels from Fig. 9b it appears to me that because the region with highest warming rates in the GER-3 simulations is so near to the western edge of the CPM modelling domain, and according to the EUR-12 simulations this region is actually extending further west, the claim that the area with highest warming rates shifts to the east can, instead of being a genuine outcome, simply be an artefact induced by the proximity of the lateral boundary zone.

Additionally, a further complication is that the role of internal variability can not be assessed here, because the authors have limited their downscaling experiments to one member based on the presumption that the ensemble spread in the GCM-experiments appeared is small enough during the heat wave episode. But without testing it is hard to make it plausible that this is a justifiable assumption.

For the authors interest, the results from de Vries et al. (2024) do not indicate an eastward shift of the area with highest warming rates relative to the area with highest temperatures in the present-day simulation (their Figs 8 and S6).

**OTHER POINTS**:

1. Line 87,146,156: Replace the word "*validation*" by "*evaluation*". A model result cannot be validated.
2. Section 2.1: I found the description somewhat confusing as if the spectral nudging is a feature of the AWI-CM1 GCM. I think though it should be considered an extension to enforce simulations in free climate mode to be constrained by ERA5-reanalyses. I suggest to remove the two words "spectral nudged" from line 92, and rephrase line

100 as "In the storyline experiments the evolution of the AWI-CM1 large-scale atmospheric circulation is constrained by spectrally nudging the model vorticity and divergence …"

3. Line 64: "at resolutions *of less* than 4 km" → "at resolutions *finer* than 4 km"
4. Line 120: "the spread" → "the inter-member spread"
5. Line 122-123: Please explain the meaning of R12B5 and R13B7 resolution,
6. Figure 1: Are the shown EUR-12 and GER-3 domains, the respective modelling domains including or excluding the lateral boundary zone? Please mention in the caption
7. Line 138: What does "*vn*" stand for?
8. Line 141-145: It is unclear how the soil information from ERA5 is used to adjust the forcing in the respective storyline, specifically at which stage of the model chain does it enter the computations.
9. Also the way the 4-layer temperature and soil moisture from ERA5 is mapped onto the soil mesh of either of the three models (AWI-CM1 and/or ICON-EUR-12 and/or ICON-GER-3) is unclear, in particular for soil moisture this is not a trivial mapping given the role of soil type in the ERA5 hydrological module HTESSEL. Please, clarify.
10. The remark "the temperature of the lowermost soil level (T_CL) was adjusted" sounds worrying in this context, why adapting a climate related prognostic variable, the simulation itself should keep track of an appropriate evolution.
11. Line 157: "model's output" → "model output" (and everywhere else)
12. Line 187-188: "However, as our study is focused on Central Europe, the model performance in the most western and eastern part of the domain is found to be acceptable" . According to Fig 4 and Fig S3, there is a considerable cold bias in ICON-EUR-12 maximum 2-m temperature compared to EOBS-12 with hatched areas in most of Germany, the eastern part of France, and Sweden/Norway. On the one hand, why do the authors consider the size of the bias acceptable, given the purpose of the paper, on the other hand what is the relevance of this bias, given that they are primarily interested in the heat wave response under different storylines.
13. Line 189: "indicating " is too strong phrasing, use "suggesting".
14. Figure 4: The authors might check what part of the temperature biases (right column) in ICON-EUR-12 originates from the AWI-CM1 driving fields by carrying out the RCM simulation directly driven by ERA5. Has such a simulation been conducted?
15. Lines 197-200: I recommend to omit statements like "indicating a further added value of out approach", since "added value" of high-resolution computations wrt courser-resolution has a somewhat different framing than in the context of evaluation. Also, in my perception the result from ICON-GER-3 wrt ICON-EUR-12 yields only an improvement for maximum 2-m temperature, and not for mean and 2-meter minimum temperature.
16. Line 201: "improved topography" →"refined topography"
17. Figure 6: Please use a different colour indicating the 40C line, e.g. green or black.
18. Line 242: What is meant by "individual"?
19. Line 248: "… which occurs in late August when the temperature again increase during summer." Why is this happening? Please, expand on the possible mechanism behind this behaviour.

20. Line 255: "in spring and early summer." And also in September?
21. Line 260: refer to "goodness of fit" instead of R2
22. Line 268: "in line with the finding" Presumably, this is just an expression of the same finding, not an independent confirmation of it.
23. Line 271-273: Likely also because the British Isles are surrounded by sea.
24. Line 299: "to the west of it". Presumably, "to the east of it" is what is meant.
25. Line 309: avoid using "observed", but use "derived warming rate" instead
26. Line 310: "temporal"?  The authors refer to intra-seasonal variations?
27. Line 315: "extending" → "an extension of" (or "an increase of")
28. Lines 328-329: this statement is very speculative.
29. Line 336: " … we addressed here for the first time …" I am afraid this statement no longer holds, see de Vries et al. (2024) for an analysis of the heat wave of 2019, and Aalbers et al. (2023) for a comparable type of analysis of the drought episodes and heat wave of 2018.

---

## Author Comment (AC1)

We thank Referee #1 for his/her valuable comments and suggestions, which helped to improve the manuscript and to remove ambiguities/misunderstandings. Below are point-to-point responses to each comment.

GENERAL COMMENTS

The study uses spectrally nudged high-resolution regional climate model simulations to quantify the sensitivity of summer 2019 European heatwave to the thermodynamic effects of global warming. The simulations indicate that, under unchanged atmospheric circulation, the daily maximum temperatures experienced under the heatwaves increase by about 2 K for each 1 K of global mean temperature change, and locally up to 3 K to the east of the heatwave centre. This contrasts with the simulated temperatures earlier in the summer, which only increase at the same rate with the global mean temperature. These features, together with the increased diurnal temperature range during the heatwave periods, suggest a role for reduced soil moisture in the amplified temperature response.

Overall, the manuscript is interesting and well-written. I only have a few minor comments and remarks on it.

We thank Referee #1 for the positive assessment of our manuscript

DETAILED COMMENTS (substance and presentation)

1. The focus of the study should be introduced earlier. Perhaps write on L7: ... we employ an event-based storyline approach to study the 2019 summer heatwaves over Central Europe. The approach comprises three steps ....

Reply: We agree with the reviewer's suggestion, and we moved the information about the focus of the study into the suggested place:
"To isolate the more certain thermodynamic response from the less certain dynamical response to anthropogenic climate change, we employ an event-based storyline approach and focus the present study on the 2019 summer heatwaves over Central Europe. Our approach comprises three steps:"

2. "the scaling of the global mean temperature" is obscure: the global mean temperature always scales one-to-one with itself.

Reply: We agree with the reviewer that this statement could be misleading. We have modified the third research question as follows:

(3) What is the local to regional extreme temperature scaling in response to global warming for an event like the 2019 heat wave?

3. Equation (2) suggests a linear increase with height.

Reply: thank you for pointing this out. There was a mistake in the formula, which will be corrected

$$\alpha_{nudge} = B_0 \left( \frac{z - z_{start}}{z_{top} - z_{start}} \right)^2$$

4. L148-149. The ensemble spread in E-OBS characterizes the observational uncertainty in individual daily temperature fields. The observational uncertainty in (e.g.) summer mean values is expected to be considerably smaller, due to cancellation of errors whose sign varies from day to day. For this reason, comparison with the E-OBS ensemble spread understates the significance of the model-to-observation biases on longer than daily time scales.

Reply: We agree with this comment. In the revised manuscript, the method of significance estimation will be changed to the student t-test.

5. L237-238. The difference in warming rates might also relate to the difference in season (early vs. late summer), not only the extremeness of temperature.

Reply: We are grateful for pointing this out. We admit that we did not put enough focus on this aspect. It may indeed be helpful to have a month-by-month perspective. This aspect has already been discussed in Sanchez-Benitez et al. (2022) and illustrated by their Fig. S9 and Fig. 6. There, they find a gradual increase of global warming amplification from early to late summer, which indeed plays a role in the magnitude of warming rates obtained in our experiment. The increased probability of large heatwaves in late summer was also found by Hundhausen et al. (2023) and illustrated in their Fig. 9. We will add this information to the discussion section at L314.

6. L274-277. Is the mean scaling similar for all summer months, or does it increase from early to late summer following the decrease in average soil moisture?

Reply: We agree that accounting for the intraseasonal cycle of regional warming amplification would be more meaningful. As discussed in the point above, there is evidence that the mean scaling is not similar for all summer months. In the revised manuscript, we will compute mean scaling for June, July, and August separately.

7. Same as comment 2.

Reply: see reply to comment 2

8. L339-341. Is this seasonal evolution of the diurnal temperature range change specific to summer 2019, or does it also occur in the other simulated years?

Reply: We observe an increase in the diurnal temperature range in all summers during the driest periods in Central Europe. We will add a sentence on this in the manuscript.

DETAILED COMMENTS (wording and typos)

1. aimed at estimating the effect of human-induced …

2. L34-35. contributions to
3. computationally efficient
4. L70 and later. spectrally nudged
5. and 8-20 km in the Arctic
6. Caption of Figure 3, L3. give / report the RMSD to E-OBS
7. Analogously to
8. the GER-3 simulation
9. L221-222. the longitude-latitude area
10. to the east of it?

Reply: We thank you for these suggestions, and we will implement all of them in the revised manuscript

---

## Author Response (AR1)

**Referee report #1**

We thank Referee #1 for his/her valuable comments and suggestions, which helped to improve the manuscript and to remove ambiguities/misunderstandings. Below are point-to-point responses to each comment.

GENERAL COMMENTS

The study uses spectrally nudged high-resolution regional climate model simulations to quantify the sensitivity of summer 2019 European heatwave to the thermodynamic effects of global warming. The simulations indicate that, under unchanged atmospheric circulation, the daily maximum temperatures experienced under the heatwaves increase by about 2 K for each 1 K of global mean temperature change, and locally up to 3 K to the east of the heatwave centre. This contrasts with the simulated temperatures earlier in the summer, which only increase at the same rate with the global mean temperature. These features, together with the increased diurnal temperature range during the heatwave periods, suggest a role for reduced soil moisture in the amplified temperature response.

Overall, the manuscript is interesting and well-written. I only have a few minor comments and remarks on it.

We thank Referee #1  for the positive assessment of our manuscript

DETAILED COMMENTS (substance and presentation)

1. The focus of the study should be introduced earlier. Perhaps write on L7: ... we employ an event-based storyline approach to study the 2019 summer heatwaves over Central Europe. The approach comprises three steps ....

Reply: We agree with the reviewer's suggestion, and we moved the information about the focus of the study into the suggested place.

Change: see L7,8, and 13 of the marked manuscript

2. "the scaling of the global mean temperature" is obscure: the global mean temperature always scales one-to-one with itself.

Reply: We agree with the reviewer that this statement could be misleading. We have modified the third research question as follows:

(3) What is the local to regional extreme temperature scaling in response to global warming for an event like the 2019 heat wave?

Change: See L3 of the marked manuscript.

3. Equation (2) suggests a linear increase with height.

Reply: thank you for pointing this out. There was a mistake in the formula, which will be corrected.

Change: See L159 of the marked manuscript.

4. L148-149. The ensemble spread in E-OBS characterizes the observational uncertainty in individual daily temperature fields. The observational uncertainty in (e.g.) summer mean values is expected to be considerably smaller, due to cancellation of errors whose sign varies from day to day. For this reason, comparison with the E-OBS ensemble spread understates the significance of the model-to-observation biases on longer than daily time scales.

Reply: We agree with this comment. In the revised manuscript, the method of significance estimation will be changed to the student t-test.

Change: The figures containing biases were moved to the Supplementary material (see Figures S3, S4, S6, and S8 of the revised supplement).

5. L237-238. The difference in warming rates might also relate to the difference in season (early vs. late summer), not only the extremeness of temperature.

Reply: We are grateful for pointing this out. We admit that we did not put enough focus on this aspect. It may indeed be helpful to have a month-by-month perspective. This aspect has already been discussed in Sanchez-Benitez et al. (2022) and illustrated by their Fig. S9 and Fig. 6. There, they find a gradual increase of global warming amplification from early to late summer, which indeed plays a role in the magnitude of warming rates obtained in our experiment. The increased probability of large heatwaves in late summer was also found by Hundhausen et al. (2023) and illustrated in their Fig. 9.

Change: In the revised manuscript, we specified that the discussed response takes place in early summer (L270). Additionally, we completed the next paragraph with lines L281-284 of the marked manuscript

6. L274-277. Is the mean scaling similar for all summer months, or does it increase from early to late summer following the decrease in average soil moisture?

Reply: We agree that accounting for the intraseasonal cycle of regional warming amplification would be more meaningful. As discussed in the point above, there is evidence that the mean scaling is not similar for all summer months. In the revised manuscript, we will compute mean scaling for June, July, and August separately.

Change: See modified Fig. 10 and lines L323-326 of the marked manuscript

7. Same as comment 2.

Reply: see reply to comment 2

8. L339-341. Is this seasonal evolution of the diurnal temperature range change specific to summer 2019, or does it also occur in the other simulated years?

Reply: We observe an increase in the diurnal temperature range in all summers during the driest periods in Central Europe. We will add a sentence on this in the manuscript.

DETAILED COMMENTS (wording and typos)

(comments in brackets indicate the position of the changes made in the marked manuscript)

1. aimed at estimating the effect of human-induced … (see L30)
2. L34-35. contributions to (see L36)
3. computationally efficient (see L81)
4. L70 and later. spectrally nudged (changed throughout the manuscript)
5. and 8-20 km in the Arctic (see L119)
6. Caption of Figure 3, L3. give / report the RMSD to E-OBS (see page 9)
7. Analogously to (see L235)
8. the GER-3 simulation (changed)
9. L221-222. the longitude-latitude area (see L254)
10. to the east of it? (see L350)

Reply: We thank you for these suggestions. We have implemented them in the revised manuscript

**Referee report #2**

We thank Referee #2 for his/her valuable comments and suggestions, which helped to improve the manuscript and to remove ambiguities/misunderstandings. Below are point-to-point responses to each comment.

Journal: Earth System Dynamics, discussion paper esd-2024-16
Review, 27 August 2024
Title: "The European summer heatwave 2019 – a regional storyline perspective"
Authors: Tatiana Klimiuk, Patrick Ludwig, Antonio Sanchez-Benitez, Helge F. Goessling, Peter Braesicke, and Joaquim G. Pinto
Recommendation: [Major/Minor Revision]
GENERAL COMMENTS:
This article presents a storyline approach of the unfolding of European summer of 2019 on a regional scale with special focus on the heat wave end of July 2019. Next to assessing the outcomes for present-day conditions, corresponding to +1.4K global warming, the methodology is repeated to cooler pre-industrial conditions, and to warmer future climates projected at several levels of future global warming. Storylines are constructed from a GCM-RCM-CPM model chain where the GCM is spectrally nudged to ERA5 vorticity and divergence to enforce the GCM atmospheric circulation to stay close to observed circulation patterns. GCM states are subsequently used for downscaling at 12 km for the European domain and 3 km for a Central-European region encompassing Germany. A performance assessment of the present-day climate simulation indicates an improvement of representing 2-meter temperature by the RCM and even more so by the CPM compared to the GCM. A primary finding of the storyline approach is a doubling, and at some locations, almost a tripling of local warming rates relative to the background warming during the episode prior

and during the heat wave whereas earlier in the season (late spring/early summer) this ratio tend to be much smaller.

The storyline perspective built from constraining the climate model state through spectral nudging to a quasi-observed state (followed by standard dynamical downscaling) provides an elegant method to isolate the thermodynamic response to anthropogenically induced climate change from the circulation response. That part of the work is already developed and described in the paper by Sanchez-Benitez et al. (2022).

The present article focuses on the subsequent downscaling steps and potential impact on a regional to local scale. Overall it is well written and of general interest, however a number of issues require attention before the manuscript is suitable for publication.

We thank Referee #2 for the critical and detailed assessment of our manuscript.

MAJOR POINTS:

1. What I found quite surprising to notice is that the authors seem not aware, at least nowhere in the manuscript any reference is made, of a widely used alternative approach, commonly referred to as Pseudo Global Warming (PGW), in which following a comparable methodology storylines are built from primarily the thermodynamic (or physical) responses to projected future global warming.

Examples in literature include for example:

• Schär et al., 1996 Surrogate climate-change scenarios for regional climate models Geophys. Res.
Lett. 23 669–72

• Prein et al., 2017.: The future intensification of hourly precipitation extremes, Nat. Clim. Change,
7, 48–52, https://doi.org/10.1038/nclimate3168

• Aalbers et al., 2023 The 2018 West-Central European drought projected in a warmer climate: how much drier can it get? Nat. Hazards Earth Syst. Sci. 23 1921–46
https://doi.org/10.5194/nhess-23-1921-2023

• Brogli et al., 2023 The pseudo-global-warming (PGW) approach: methodology, software package PGW4ERA5 v1.1, validation and sensitivity analyses Geosci. Model Dev. 16 907–26 https://doi.org/10.5194/gmd-16-907-2023

Moreover, just two weeks after this manuscript was submitted a paper by H. de Vries et al., entitled "Western Europe's extreme July 2019 heatwave in a warmer world", appeared in Environmental Research: Climate ( https://doi.org/10.1088/2752-5295/ad519f ) in which the authors develop a storyline perspective built on the PGW approach focusing on the very same event.

In my opinion, reference to the PGW-approach in the context of the application presented in this paper is required, and I would strongly encourage to include a discussion on the pro's and cons of applying the respective methods (spectral nudging and PGW) in this type of storyline development, focusing on events or episodes.

Reply: We agree that we have not discussed this approach in the previous version of this manuscript, and this will be changed in the new version of the paper. We are very aware of the PGW approach, and indeed, we have used it in other studies (e.g. Ludwig et al. 2023 focusing on the Central European floods 2021, https://doi.org/10.5194/nhess-23-1287-2023).

After a brief introduction to both approaches, we will mention that the usage of a nudged global climate model was motivated by the absence of a need to make any assumptions on the delta fields, creating physically consistent dynamical and thermodynamical conditions that would correspond to the synoptic evolution of that particular event, including the SSTs, as the AWI-CM1 is a coupled climate model. On the contrary, the PGW approach does not account for the interannual variability of those fields (Brogli et al., 2023).

On the other hand, one of the advantages of the PGW approach over ours is the potential to avoid GCM-specific biases by repeating the experiment with deltas derived from various models or model means. In our work, we follow the path prescribed by a single GCM. In the context of the storyline approach, this unfolding of events is physically self-consistent and plausible, which complies with the definition of a storyline introduced by Shepherd (2018) and allows for a process-oriented evaluation of the obtained responses. This text will be expanded and included in the manuscript. We will, of course, also cite de Vries et al., 2024, which we were not aware of.

Change: We have modified the introduction in accordance with the provided reply. See L50, and L56 - L72 in the marked document

2. It is unfortunate that the region with highest temperatures during the heat wave episode is on the western edge of the ICON-GER-3 domain (see Figure 6). In particular, it makes the conclusion that the region of highest relative warming rates shifts to the east questionable. Comparing both panels from Fig. 9b it appears to me that because the region with highest warming rates in the GER-3 simulations is so near to the western edge of the CPM modelling domain, and according to the EUR-12 simulations this region is actually extending further west, the claim that the area with highest warming rates shifts to the east can, instead of being a genuine outcome, simply be an artefact induced by the proximity of the lateral boundary zone.

Reply: We agree that the GER-3 domain does not cover central parts of France, where the peak temperatures occurred. This is due to the specificity of the Innopool SCENIC project, which focuses on extreme events in Germany and associated impacts.

As you mentioned, there is almost no shift of the heatwave's core: in Figure 6i (+4K), we see that the 45°C contour occupies a similar region occupied by the 42°C contour in Figure 6c (present-day). However, with our analysis, we want to convey that the areas with the highest warming rates are found outside the heatwave's core region in both EUR-12 and GER-3 simulations. This can be illustrated by Figure S5 of our supplementary material and with an additional Figure R1, presented below, which shows specifically the warming rates. In Figures S5 and R1, the core of the heatwave is outlined by contour lines, and the temperature difference of the +4K simulation to the present day is shown by shading. Both EUR-12 and GER-3 (see Figure S5) simulations show that the maximum temperature delta is outside of the 42°C contour. But indeed, the warming rates within the heatwave's core are still high (see Figure R1).

Thus, we would like to keep the simulation domains as they are because, with our experiment setup, the largest warming rates are captured within the domain. However, we understand the reviewer's concern. Therefore, we have added a statement that different

choices could have been made if the focus of the project had been different. We also plan to add Figure R1 to the supplementary material.

[Figure]

**Figure R1.** Ensemble mean of warming rates for the 25th of July 2019. The black contour delineates the 42°C threshold; the grey box indicates the boundaries of the GER-3 domain.

Changes: Figure S16 of the revised supplementary material shows ensemble mean warming rates for the target periods. We have commented on the topics discussed above in lines L310-312, as well as in lines L378-L382 of the discussion section.

Additionally, a further complication is that the role of internal variability can not be assessed here, because the authors have limited their downscaling experiments to one member based on the presumption that the ensemble spread in the GCM-experiments appeared is small enough during the heat wave episode. But without testing it is hard to make it plausible that this is a justifiable assumption.

Reply: We acknowledge the importance of utilising all five available AWI-CM1 ensemble members to enhance the robustness of our results. At the time of manuscript preparation, we were technically limited to processing only ensemble member 1. With the entire ensemble now available for the EUR-12 domain, we will incorporate the uncertainty ranges into Figures 3, 7, and 8 (see an example of changed Figures 7a and 8d in Figures R2 and R3 below). Additionally, we will provide the ensemble range of the focus period of the 23-27th of July and the 25th of July in the supplementary materials. We would like to keep the 2d plots as they are to be consistent with the GER-3 simulation, which is based on the ensemble member 1.

[Figure]

**Figure R2.** Daily maximum temperatures averaged over the longitude/latitude box with boundaries 48° N 51° N and 6° E - 10° E (see Fig. 1b) over the MJJAS period of the year 2019 based on the EUR-12 storyline simulations. Shading spans the minimum/maximum range of values obtained from the five-member ensembles.

[Figure]

**Figure R3.** Ensemble mean of warming rates for the running mean (5-day window) of daily maximum, mean, and minimum temperatures over the same box as in Figure R2. Shading spans the minimum/maximum range of values obtained from the five-member ensembles.

Changes:

- We have added the available ensemble ranges to Fig. 3, Fig.7, and Fig. 8
- Fig. 4 is now based on all ensemble members
- Ensemble spread is shown in Figures S10 and S11 of the revised supplementary material
- The text was respectively changed in L147-149 (lines as of the marked manuscript)

For the authors interest, the results from de Vries et al. (2024) do not indicate an eastward shift of the area with highest warming rates relative to the area with highest temperatures in the present-day simulation (their Figs 8 and S6).

Reply: We thank the Reviewer for suggesting that such a discussion be added to the paper, as this addition would strengthen the manuscript and allow a more comprehensive view of the topic. We have carefully read the work by de Vries et al. (2024) and found that the warming rates in Central France shown in their Figure S6 are similar to ours and range from 1.5 to 2.5 °C/K (see Figure R1). Indeed, the areas with warming rates exceeding 2.5 °C/K

are different from de Vries et al. (2024), but in both studies, they are outside of the core region of the French heatwave. So, we do not think that their results and ours contradict each other, even though the approaches are quite different. Thus, we argue that both studies provide a different perspective on how the 2019 heat wave would develop in a warmer world. We will add the above-mentioned points and other points to the discussion part of our manuscript.

Changes: We have extended the discussion section with the lines L392-400 of the marked manuscript

OTHER POINTS:

1. Line 87,146,156: Replace the word "validation" by "evaluation". A model result cannot be validated.

Reply: Thanks, this will be implemented
Changes: See L173 and L183 of the marked manuscript

2. Section 2.1: I found the description somewhat confusing as if the spectral nudging is a feature of the AWI-CM1 GCM. I think though it should be considered an extension to enforce simulations in free climate mode to be constrained by ERA5-reanalyses. I suggest to remove the two words "spectral nudged" from line 92, and rephrase line 100 as "In the storyline experiments the evolution of the AWI-CM1 large-scale atmospheric circulation is constrained by spectrally nudging the model vorticity and divergence …"

Reply: We agree with rephrasing the line 100 in a suggested way. We will also consider removing "spectral nudged" from line 92.

Change: see L121

3. Line 64: "at resolutions of less than 4 km" →"at resolutions finer than 4 km"

4. Line 120: "the spread" → "the inter-member spread"
5. Line 122-123: Please explain the meaning of R12B5 and R13B7 resolution,
Reply: Thanks, all of the above will be implemented
Changes: see L83, L142, and L145 of the marked manuscript

6. Figure 1: Are the shown EUR-12 and GER-3 domains, the respective modelling domains including or excluding the lateral boundary zone? Please mention in the caption

Reply: The domains shown in Figure 1 include the lateral boundary zone. We will mention it in the caption.

Change: See caption to Fig. 1

7. Line 138: What does "vn" stand for?

Reply: Thank you for pointing this out. *vn* is not used in any equation. Thus, we will omit this in line 138. But generally, *vn* stands for the velocity normal to the edge of the triangular grid cell.
Change: see L162

8. Line 141-145: It is unclear how the soil information from ERA5 is used to adjust the forcing in the respective storyline, specifically at which stage of the model chain does it enter the computations.

Reply: Sorry for this inconsistency. We will mention in the text that the soil temperature and soil moisture from ERA5 were used to initialise the EUR-12 simulations due to the partial unavailability of soil temperature in AWI-CM1 outputs. To let the soil adapt to different climate conditions, we used a longer spin-up time for storylines, as described in L142-143.

Change: see L165-166

9. Also the way the 4-layer temperature and soil moisture from ERA5 is mapped onto the soil mesh of either of the three models (AWI-CM1 and/or ICON-EUR-12 and/or ICON-GER-3) is unclear, in particular for soil moisture this is not a trivial mapping given the role of soil type in the ERA5 hydrological module HTESSEL. Please, Clarify.

Reply: The soil temperature and moisture entering ICON in their original form are being pre-processed by the built-in algorithm. To account for the possible discrepancy of soil types in ICON and ERA5, the volumetric soil moisture is transformed into the universal soil moisture index (SMI), which takes into account different soil types (see e.g.. Prill et al. 2023, DOI: 10.5676/DWD_pub/nwv/icon_tutorial2023).

10. The remark "the temperature of the lowermost soil level (T_CL) was adjusted" sounds worrying in this context, why adapting a climate related prognostic variable, the simulation itself should keep track of an appropriate evolution.

Reply: We agree that this sentence looks misleading. This is a technical detail of the soil layer treatment in the model. In ICON, the lowermost soil moisture level is not prognostic but serves as a lower boundary condition for the heat conduction equation (Schulz et al., 2016, DOI: 10.1127/metz/2016/0537). It is, by default, set to the climatological annual mean near-surface temperature T_CL based on the Climate Research Unit data (Mitchel and Jones, 2005, https://doi.org/10.1002/joc.1181). We adjusted this lower boundary condition by adding (or subtracting, in case of pre-industrial climate) 1°C to it for each corresponding storyline.

Change: see L168-170

11. Line 157: "model's output" → "model output" (and everywhere else)

Reply: Thanks, this will be implemented
Change: see L184

12. Line 187-188: "However, as our study is focused on Central Europe, the model performance in the most western and eastern part of the domain is found to be acceptable" . According to Fig 4 and Fig S3, there is a considerable cold bias in ICON-EUR-12 maximum 2-m temperature compared to EOBS-12 with hatched areas in most of Germany, the eastern part of France, and Sweden/Norway. On the one hand, why do the authors consider the size of the bias acceptable, given the purpose of the paper, on the other hand what is the relevance of this bias, given that they are primarily interested in the heat wave response under different storylines.

Reply: Thank you for pointing this out. Essentially, the information about the bias is there only to illustrate that the underestimation of daily maximum temperature found in AWI-CM1 output was reduced over Central Europe by dynamical downscaling. We admit that the fact that most of Central Europe is hatched in Fig. 4c puts into question the relevance of this information for the study. Moreover, the time series in Fig. 3a, as well as the delta-RMSD maps in Fig. 4b show a clear improvement in TMAX representation with dynamical downscaling. We are considering removing the right columns (bias maps) from Fig. 4 and 5 and moving them to the supplementary material, adding the bias maps of the driving AWI-CM1 simulation and control run. Additionally, the direct comparison of seasonally averaged temperature fields is already given in Fig. S2 of the supplementary. In Figure R4 below, one can see the reduction in bias in central Europe.

[Figure]

**Figure R4.** seasonal mean bias of daily maximum temperature with respect to E-OBS; left: AWI-CM1, right: ICON-EUR-12

Change: The bias maps were completed with the results for AWI-CM1 and the control run and moved to the supplementary: see Figures S3, S4 of the revised supplementary material

13. Line 189: "indicating " is too strong phrasing, use "suggesting".

Reply: Thanks, this will be implemented
Change: see L220

14. Figure 4: The authors might check what part of the temperature biases (right column) in ICON-EUR-12 originates from the AWI-CM1 driving fields by carrying out the RCM simulation directly driven by ERA5. Has such a simulation been conducted?

Reply: Thank you for this comment. We conducted the control simulation directly driven by ERA5 but did not include it in the manuscript. The control run bias maps for daily maximum,

mean, and minimum temperatures over the JJA period are shown in Figure R5. As the patterns in Figures R5 (left) and R4 (right) are similar, we assume that the bias of AWI-CM1 did not propagate strongly into the regional simulations. We will add to the manuscript the part on the contribution of AWI-CM1 driving fields and ICON-EUR-12 downscaling into the summer temperature biases.

[Figure]

**Figure R5.** seasonal mean bias of daily maximum (left), mean (middle), and minimum (right) temperature with respect to E-OBS;

Change: See Figures S3, S4 of the revised supplementary material

15. Lines 197-200: I recommend to omit statements like "indicating a further added value of out approach", since "added value" of high-resolution computations wrt courser resolution has a somewhat different framing than in the context of evaluation. Also, in my perception the result from ICON-GER-3 wrt ICON-EUR-12 yields only an improvement for maximum 2-m temperature, and not for mean and 2-meter minimum Temperature.

Reply: we will remove this part

Change: see L229

16. Line 201: "improved topography" →"refined topography"
Change: see L233

17. Figure 6: Please use a different colour indicating the 40C line, e.g. green or black.

Reply: Thanks, this will be implemented

Change: see Figure 6 of the revised manuscript

18. Line 242: What is meant by "individual"?

Reply: We agree that the word "individual" is misleading. We meant TMAX, TMIN, and TMEAN. It will be changed in the revised manuscript.

Change: see L274

19. Line 248: "… which occurs in late August when the temperature again increase during summer." Why is this happening? Please, expand on the possible mechanism behind this behaviour.

Reply: Thank you for the comment. We admit that no discussion of the last warming rate peak has been provided. We plan to comment on the tendency of warming rates to increase towards late summer, which presumably happens due to the decrease in soil moisture. The high warming rates of the late August heat wave might be explained by this intraseasonal dependency (see, e.g. Hundhausen et al. (2023)).

Change: we have expanded the paragraph with lines L281-284 in the revised manuscript

20. Line 255: "in spring and early summer." And also in September?

Reply: Thank you. We will also comment on the alignment of TMIN and TMAX warming rates by the end of the hot season.

Change: see L294

21. Line 260: refer to "goodness of fit" instead of R2

Reply: Thanks, this will be implemented

Change: See L298

22. Line 268: "in line with the finding" Presumably, this is just an expression of the same finding, not an independent confirmation of it.

Reply: We agree with the comment. This expression will be replaced by "underscoring"

Change: see L309

23. Line 271-273: Likely also because the British Isles are surrounded by sea.

Reply: Thank you. We will revise and extend this sentence.

Change: see L316

24. Line 299: "to the west of it". Presumably, "to the east of it" is what is meant.

Reply: indeed, we will change this.

Change: see L350

25. Line 309: avoid using "observed", but use "derived warming rate" instead

Reply: Thanks, this will be implemented

Change: see L362

26. Line 310: "temporal"? The authors refer to intra-seasonal variations?

Reply: yes, one can also call them intra-seasonal.

Change: see L362

27. Line 315: "extending" → "an extension of" (or "an increase of")

Reply: Thanks, this will be implemented

Change: see L367

28. Lines 328-329: this statement is very speculative.

Reply: The sentence will be changed as follows:

The high but limited (<2.5 on the 25th of July) warming rates at the centre of the July 2019 heatwave may be explained by the possible decrease in the strength of the response of the soil moisture-temperature coupling over desiccated soils (e.g., Gevaert et al., 2018)

Change: We replaced the original sentence by a slightly weaker statement (see L386)

29. Line 336: " … we addressed here for the first time …" I am afraid this statement no longer holds, see de Vries et al. (2024) for an analysis of the heat wave of 2019, and Aalbers et al. (2023) for a comparable type of analysis of the drought episodes and heat wave of 2018.

Reply: We agree that the temporal evolution of a heat wave in several alternative climates on a regional scale has been investigated in the mentioned works. We will reformulate this sentence, emphasising that the specificity of our approach is in the downscaling of global nudged storylines obtained with the coupled climate model.

Change: see L403

---

## Referee Report (RR1)

Journal: Earth System Dynamics, discussion paper esd-2024-16
Review Revised manuscript, 22 November 2024

Title: "The European summer heatwave 2019 – a regional storyline perspective"

Authors: Tatiana Klimiuk, Patrick Ludwig, Antonio Sanchez-Benitez, Helge F. Goessling, Peter Braesicke, and Joaquim G. Pinto

Recommendation: [Minor Revision]

**GENERAL**:

I want to thank the authors for their detailed and extensive reply to the points I raised when reviewing the original manuscript. As far as I am concerned, the authors have adequately addressed most of the issues, leaving just a few points that still require attention, before the manuscript is suitable for publication.

**REMAINING POINTS**:

1. In reply to my previous point 15 regarding the, in my opinion, improper use of the wording "added value" the authors have removed one "added value" phrase, but retained the rest. In my perception, the reduction in temperature bias when going from the GCM to the RCM is merely the outcome of a using different physics formulations, and also because the GCM apparently performs very poorly for near-surface temperature. I don't think the higher resolution itself plays a role in the improved skill of the near-surface temperature representation. Unless the authors can make plausible the bias reduction is caused by the higher resolution of the RCM wrt the GCM, I advise to avoid the wording "added value", and use "improved", "gain", "benefit" or comparable wording.

2. Lines 358-359 (the authors reply to my previous point 28): This is a possible explanation but without testing it remains highly speculative. The 2019 heat wave event was relatively short-lasting and soil conditions prior to the event were not particularly dry (unlike in 2018). In my opinion it is at least equally plausible that the region with highest future warming is slightly displaced with respect to the region with highest temperature in the reference run, because the overlying atmospheric flow pattern with highest temperatures in the future runs is slightly off. It is just very unlikely that the region with highest temperatures in the future runs precisely collocates with that region in the control run, and, thus, highest warming rates will always be found off the centre with highest temperatures in the control run.

    Please express that, without testing, the explanation you provide is highly speculative, and that alternative explanations (or combinations of them) are equally plausible.

**OTHER POINTS**:

1. Line 65:"from various models or model means". This is a somewhat vague formulation. In practice I would say the deltas are derived "from multi-model ensemble means or single-model multi-member ensemble means" (see de Vries et al. for examples).
2. Line 78: CPM stands for Convection Permitting Models (and not Convective Permitting Models). Please adjust here and throughout the remainder of the text.
3. Line 135/136 (and further down): The word *dynamical* in the phrase "the dynamical year" is somewhat confusing, although I presume it stands for "large-scale dynamical constraints inferred from ERA5". For clarity, please, make explicit what you mean at the first occurrence.
4. Line 135:".. the 31$^{st}$ of September .." → either ".. the 30$^{th}$ of September .." or ".. the 31$^{st}$ of December"
5. Line 156-157: I could not find a reference of your reply to my previous point 9 regarding the mapping of soil information from ERA5 to ICON in the revised manuscript. Please mention explicitly it in the text, at least including the reference to Prill et al.
6. Lines 159-160: Change "lowermost soil *level*" into "lowermost soil *layer*" (or "bottom soil *layer*")
7. Lines 159-162: Regarding your reply to my previous point 10 (on the meaning of T_CL) I am wondering if there is a similar treatment of bottom layer soil moisture. Could you spend one or two lines on that?
8. Line 276: "… Autumn" → *"early autumn"* (simulations stop end of September); I also suggest to change "… other summers …" by "… the other summer seasons (May-September) …" (or (MJJAS))
9. Line 355: "should cover" →"should have covered"
10. Line 355: "Nevertheless, …"→"Still, …"
11. Caption Figures 3, 7, 8: "five-member" → "5-member"
12. Captions Figures S3 and S4: "a-d" should be "a-c"

---

## Author Response (AR2)

**Authors response**

Journal: Earth System Dynamics, discussion paper esd-2024-16
Review Revised manuscript, 22 November 2024
Title: "The European summer heatwave 2019 – a regional storyline perspective"
Authors: Tatiana Klimiuk, Patrick Ludwig, Antonio Sanchez-Benitez, Helge F.
Goessling, Peter Braesicke, and Joaquim G. Pinto
Recommendation: [Minor Revision]

GENERAL:
I want to thank the authors for their detailed and extensive reply to the points I raised
when reviewing the original manuscript. As far as I am concerned, the authors have
adequately addressed most of the issues, leaving just a few points that still require
attention, before the manuscript is suitable for publication.

We thank Referee #2 for the valuable comments. We were glad to elaborate on the raised
issues and have addressed the remaining points as follows:

REMAINING POINTS:
1. In reply to my previous point 15 regarding the, in my opinion, improper use of the wording
"added value" the authors have removed one "added value" phrase, but retained the rest. In
my perception, the reduction in temperature bias when going from the GCM to the RCM is
merely the outcome of a using different physics formulations, and also because the GCM
apparently performs very poorly for near-surface temperature. I don't think the higher
resolution itself plays a role in the improved skill of the near-surface temperature
representation. Unless the authors can make plausible the bias reduction is caused by the
higher resolution of the RCM wrt the GCM, I advise to avoid the wording "added value", and
use "improved", "gain", "benefit" or comparable wording.

Reply: Thank you for the comment. We agree that in the context of our study, it is more
appropriate to refer to  "improvement" for near-surface temperature representation by
dynamical downscaling rather than using the more specific concept of "added value". In
general terms, the added value of RCMs compared to GCMs is primarily due to a better
representation of the physical processes, the orography and the surface characteristics, thus
facilitating an improved perspective of the atmospheric circulation and its impacts at regional
to smaller scales (rainfall, temperature, wind, and others) and thus closer to the
observations.

The following sentences were rephrased in the manuscript:

L13/14: We provide evidence that the downscaling of global storyline integrations
significantly *improved the representation of* present-day temperature patterns and reduced
error in daily 2m temperature relative to observations in Central Europe.

L97/98: (1) How accurately can a regional event-based storyline simulation represent a
recent event, and what is the *improvement* compared to the global spectrally nudged
storyline simulation

L176/177: The root mean square difference (RMSD) to observational datasets (DWD and E-OBS) and its change between simulations of different resolutions (ΔRMSD) is chosen as a metric to *compare the representation of near-surface temperature by the models of our GCM-RCM-CPM chain in the present-day storyline.*

L192/193: *We compared the performance of the simulations within the model chain by calculating the* root mean square difference (RMSD) in June - August of the simulated 2m temperature with respect to DWD observations …

L203: *We* interpolated the ICON EUR-12 data to the grid of AWI-CM1 and compared the RMSD of both models to E-OBS.

L216: The nested convection-permitting GER-3 simulation was assessed by comparing the RMSD of the 2m temperature to observations with the RMSD of the driving EUR-12 simulation.

Caption of Figures 4 and 5: Performance assessment …

L228: Analogously to the EUR-12 simulation, the *evaluation of* GER-3 simulation…

L233: Given the *improved performance* by dynamical downscaling with ICON-CLM for present-day conditions,...

L325: (1) How accurately can a regional event-based storyline simulation represent a recent event, and what is *the improvement* compared to the global spectrally nudged storyline simulation

2. Lines 358-359 (the authors reply to my previous point 28): This is a possible explanation but without testing it remains highly speculative. The 2019 heat wave event was relatively short-lasting and soil conditions prior to the event were not particularly dry (unlike in 2018). In my opinion it is at least equally plausible that the region with highest future warming is slightly displaced with respect to the region with highest temperature in the reference run, because the overlying atmospheric flow pattern with highest temperatures in the future runs is slightly off. It is just very unlikely that the region with highest temperatures in the future runs precisely collocates with that region in the control run, and, thus, highest warming rates will always be found off the centre with highest temperatures in the control run. Please express that, without testing, the explanation you provide is highly speculative, and that alternative explanations (or combinations of them) are equally plausible.

Reply: We understand the concern of the reviewer and have modified the paragraph respectively.

In the marked-up version of the manuscript, lines L369-380 include the following changes:

*Several factors may have influenced the spatially variable magnitude of the 2m temperature response, which is seen both for a single member (Fig. 9) as well as for all ensemble members (Fig. S16). One possible explanation for this behaviour is a heterogeneous*

*response of soil moisture-temperature coupling within and outside the heatwave's core (see e.g., Gevaert et al., 2018; Miralles et al., 2014). We hypothesise that the overall amplification of the warming rates during heatwave events and the extension of the diurnal temperature range may have been exacerbated due to soil-atmosphere feedback. However, a dedicated and detailed analysis would be necessary to demonstrate or dismiss this hypothesis, as other factors like small changes in atmospheric dynamics may have also played an important role.*

OTHER POINTS:
Reply: Thank you for the thorough review of the text. We have elaborated on the points raised below and provide here the numbers of the lines in the marked-up *version* of the manuscript, where you can find the changes applied

1. Line 65:"from various models or model means". This is a somewhat vague formulation. In practice I would say the deltas are derived "from multi-model ensemble means or single-model multi-member ensemble means" (see de Vries et al. for examples).

Change in L66/67: On the other hand, one of the advantages of the PGW approach over the nudged storyline approach is the potential to avoid GCM-specific biases by repeating the experiment with deltas derived *from multi-model ensemble means or different single-model multi-member ensemble means (see e.g., Aalbers et al., 2023; Vries et al., 2024)*

2. Line 78: CPM stands for Convection Permitting Models (and not Convective Permitting Models). Please adjust here and throughout the remainder of the text.

Change: L80, L99, L216, caption of Figure 5, L332

3. Line 135/136 (and further down): The word dynamical in the phrase "the dynamical year" is somewhat confusing, although I presume it stands for "large-scale dynamical constraints inferred from ERA5". For clarity, please, make explicit what you mean at the first occurrence.

Change in L131/132: *Throughout the text, for all storylines, we refer to the years corresponding to the present-day circulation inferred from ERA5 as the "dynamical years" 2017–2022.*

4. Line 135:".. the 31st of September .." → either ".. the 30th of September .." or ".. the 31st of December"
Change in L130: Each storyline is simulated continuously from the 1st of January 2017 to the 30th of September 2022.

5. Line 156-157: I could not find a reference of your reply to my previous point 9 regarding the mapping of soil information from ERA5 to ICON in the revised manuscript. Please mention explicitly it in the text, at least including the reference to Prill et al.

Change in L160-163: *In ICON, the initialising soil data is pre-processed and remapped onto the 8-layer mesh by the built-in algorithm (Prill et al., 2023; Pham et al., 2021). To account for the possible discrepancy of soil types between ICON and ERA5, the volumetric soil*

*moisture is transformed into the universal soil moisture index (SMI), which makes it independent of the soil type (Prill et al., 2023).*

6. Lines 159-160: Change "lowermost soil level" into "lowermost soil layer" (or "bottom soil layer")

Change in L166: Additionally, the temperature of the *bottom soil layer,* which is not prognostic *in the TERRA land module of ICON* but is set to the climatological annual mean near-surface temperature T\_CL based on the Climate Research Unit data,…

7. Lines 159-162: Regarding your reply to my previous point 10 (on the meaning of T_CL) I am wondering if there is a similar treatment of bottom layer soil moisture. Could you spend one or two lines on that?

Change in L169-170: *The lower boundary condition for soil moisture is given by a free-drainage formulation and thus did not require additional adjustments (Prill et al., 2023; Chen et al., 2018; Zeng and Decker, 2009).*

8. Line 276: "... Autumn" → "early autumn" (simulations stop end of September); I also suggest to change "... other summers ..." by "... the other summer seasons (May-September) ..." (or (MJJAS))

Change in L286: In contrast, the temperature response appears to be distributed more uniformly during the day in spring, early summer, and *early autumn*. We obtained similar behaviour in other *summer seasons (May - September)* of the simulated period.

9. Line 355: "should cover" →"should have covered"
10. Line 355: "Nevertheless, ..."→"Still, ..."

Change in L366: We admit that for a more comprehensive km-scale investigation of this heatwave, this domain should *have covered* a larger fraction of France. *Still*, as this study is part of the Innopool SCENIC project…

11. Caption Figures 3, 7, 8: "five-member" → "5-member"
Change: see captions of Figures 3, 7, 8

12. Captions Figures S3 and S4: "a-d" should be "a-c"
Change: see captions of figures S3 and S4